# Global distribution patterns of marine nitrogen-fixers by imaging and molecular methods

Juan José Pierella Karlusich[1,2], Eric Pelletier[2,3], Fabien Lombard [2,4,5], Madeline Carsique[1], Etienne Dvorak [1], Sébastien Colin[6,7,8], Marc Picheral[2,4], Francisco M. Cornejo-Castillo [7], Silvia G. Acinas [9], Rainer Pepperkok[2,6], Eric Karsenti[1,2,6], Colomban de Vargas[2,7], Patrick Wincker [2,3], Chris Bowler[1,2✉] & Rachel A. Foster [10✉]

Nitrogen fixation has a critical role in marine primary production, yet our understanding of marine nitrogen-fixers (diazotrophs) is hindered by limited observations. Here, we report a quantitative image analysis pipeline combined with mapping of molecular markers for mining >2,000,000 images and >1300 metagenomes from surface, deep chlorophyll maximum and mesopelagic seawater samples across 6 size fractions (<0.2–2000 μm). We use this approach to characterise the diversity, abundance, biovolume and distribution of symbiotic, colony-forming and particle-associated diazotrophs at a global scale. We show that imaging and PCR-free molecular data are congruent. Sequence reads indicate diazotrophs are detected from the ultrasmall bacterioplankton (<0.2 μm) to mesoplankton (180–2000 μm) communities, while images predict numerous symbiotic and colony-forming diazotrophs (>20 μm). Using imaging and molecular data, we estimate that polyploidy can substantially affect gene abundances of symbiotic versus colony-forming diazotrophs. Our results support the canonical view that larger diazotrophs (>10 μm) dominate the tropical belts, while unicellular cyanobacterial and non-cyanobacterial diazotrophs are globally distributed in surface and mesopelagic layers. We describe co-occurring diazotrophic lineages of different lifestyles and identify high-density regions of diazotrophs in the global ocean. Overall, we provide an update of marine diazotroph biogeographical diversity and present a new bioimaging-bioinformatic workflow.

[1] Institut de Biologie de l'ENS (IBENS), Département de biologie, École normale supérieure, CNRS, INSERM, Université PSL, Paris, France. [2] CNRS Research Federation for the study of Global Ocean Systems Ecology and Evolution, FR2022/Tara Oceans GOSEE, Paris, France. [3] Génomique Métabolique, Genoscope, Institut François Jacob, CEA, CNRS, Univ Evry, Université Paris-Saclay, Evry, France. [4] Sorbonne Universités, CNRS, Laboratoire d'Océanographie de Villefranche (LOV), Villefranche-sur-Mer, France. [5] Institut Universitaire de France (IUF), Paris, France. [6] European Molecular Biology Laboratory, Heidelberg, Germany. [7] Sorbonne Université, CNRS, Station Biologique de Roscoff, UMR 7144, ECOMAP, Roscoff, France. [8] Max Planck Institute for Developmental Biology, Tübingen, Germany. [9] Department of Marine Biology and Oceanography, Institut de Ciènces del Mar, CSIC, Barcelona, Spain. [10] Department of Ecology, Environment and Plant Sciences, Stockholm University, Stockholm, Sweden. ✉email: cbowler@bio.ens.psl.eu; rachel.foster@su.se

Approximately half of global primary production occurs in the oceans[1], fueling marine food webs, plankton decomposition and sequestration of fixed carbon to the ocean interior. Marine primary production is often limited by nitrogen (N) in vast expanses of the open ocean (~75% of surface ocean)[2,3]. In these regions, the biological reduction of di-nitrogen gas ($N_2$) to bioavailable N, a process called biological $N_2$ fixation (BNF), is a critical source of new N to the ecosystem and ultimately controls the uptake and sequestration of carbon dioxide ($CO_2$) on geologic time scales[4–6].

In the upper sunlit ocean, it was traditionally thought that BNF was largely restricted to the subtropical and tropical gyres and mediated by a few groups of larger sized $N_2$-fixing (diazotrophic) cyanobacteria and holobionts (>10 μm): the colony-forming non-heterocystous *Trichodesmium* spp., and heterocystous cyanobacterial (*Richelia intracellularis* and *Calothrix rhizosoleniae*, hereafter *Richelia* and *Calothrix*, respectively) symbionts of diatoms (diatom-diazotroph associations; DDAs)[7]. More recently, unicellular cyanobacteria (UCYN) have been detected in environmental samples outside the tropical belts by qPCR targeting the BNF marker gene *nifH*[8]. One of these $N_2$-fixing UCYN groups is *Candidatus* Atelocyanobacterium thalassa (hereafter UCYN-A). Three UCYN-A lineages (A-1, A-2, A-3) live in symbiosis with a small single celled eukaryote (haptophyte)[9–11]. UCYN-B is another unicellular group that is most closely related to *Crocosphaera watsonii* (hereafter *Crocosphaera*). UCYN-B lives singly, colonially or in symbioses with a large chain-forming diatom (*Climacodium frauenfeldianum*)[12–14]. UCYN-C is the third marine unicellular group identified thus far by *nifH* sequence, and is most closely related to the free-living unicellular diazotroph *Cyanothece sp.* ATCC 51142[15], and less studied. Finally, non-cyanobacterial diazotrophs (NCDs), including Archaeal and Bacterial lineages, co-occur with the cyanobacterial diazotrophs in the surface ocean and additionally below the photic layer. The distribution and in situ activity of NCDs are poorly constrained and difficult to estimate[16–18].

The first global ocean database of diazotrophs was compiled for the MARine Ecosystem DATa (MAREDAT) project[19]. It includes cell counts for diazotrophs that can be identified by microscopy (*Trichodesmium*, *Richelia*, and *Calothrix*), as well as *nifH* qPCR datasets which additionally cover UCYN-A, UCYN-B, and UCYN-C (Supplementary Fig. S1a). A recent update of the MAREDAT dataset resulted in more than doubling of the *nifH* observations[20]. However, both MAREDAT and the updated version still have low coverage in vast regions of the global ocean (Supplementary Fig. S1a). Several of these poorly sampled areas were sampled during the *Tara* Oceans circumnavigation (2009–2013)[21] (Supplementary Fig. S1b).

*Tara* Oceans collected discrete size fractions of plankton using a serial filtration system[21]; some samples were used to generate parallel molecular and imaging datasets. The *Tara* Oceans gene catalog from samples enriched in free-living prokaryotes is based on the assembly of metagenomes and is highly comprehensive[22,23]. However, the larger plankton size fractions (>0.8 μm) enriched in eukaryotes are genomically much more complex, and thus current *Tara* Oceans gene catalogs from these fractions are based only on poly-A-tailed eukaryotic RNA[24,25]. Hence, the prokaryotes from these larger size fractions have been unstudied and are limited to specific taxa based on these poly-A assembled sequences[26–28], and thus the signal is difficult to interpret quantitatively[29]. The *Tara* Oceans imaging dataset[30] is also underutilized, especially due to the lack of well-established workflows. Overall, the cyanobacterial diazotrophs, especially those with diverse lifestyles (colonial, symbiotic, chain formers), have been poorly characterized (with the exception of UCYN-A[11,16,22,26,28,31]).

Here, we report the diversity, abundance, and distribution of symbiotic, colony-forming, and particle-associated diazotrophs in the global ocean by mining >1300 metagenomes[22,24,25] and >2,000,000 high-throughput images from *Tara* Oceans. We use the single-copy core bacterial gene *recA*[32] to quantify the bacterial community in each metagenome; thus the read abundance ratio of *nifH*/*recA* provides an estimate for the relative contribution of diazotrophs. In parallel, we train an image classification model and utilized it with images from an Underwater Vision Profiler (UVP)[33] and confocal microscopy[30] to generate a versatile analytical pipeline from images to genomics and genomics to images. We find a remarkable congruence between the image and gene-based analyses, and several new 'hotspots' for diazotrophs are identified for the first time to our knowledge. Accordingly, diazotrophs are globally distributed and present in all size fractions, even among ultrasmall bacterioplankton (<0.22 μm). Other unexpected results include the detection of *nifH* sequences similar to freshwater obligate symbionts in multiple and distant ocean basins, a general lack of UCYN *nifH* sequences in the surface ocean, ubiquitous detection of NCDs, and a consistent co-occurrence of multiple cyanobacterial diazotrophs (from both images and genes) in several locations. Overall, this work provides an updated composite of diazotroph biogeography in the global ocean and the environmental factors that shape these patterns.

## Results and discussion

**Diazotroph abundance and biovolume based on imaging methods.** We first used machine learning tools (see "Methods"[30]) to search for diazotrophs in the *Tara* Oceans high-throughput confocal imaging dataset derived from 61 samples of the 20–180 μm plankton size fraction collected at 48 different sampling locations (Supplementary Fig. S2). More than 400 images of DDAs and almost 600 images of *Trichodesmium*-free filaments were predicted (Figs. 1 and 2; https://www.ebi.ac.uk/biostudies/studies/S-BSST529); all images were from the tropical and subtropical regions and consistent with the molecular analyses (see later). The same image recognition searches were performed on the confocal microscopy images from the 5–20 μm size fraction[30], which covers 75 samples from 51 stations. These searches detected only 8 images of short *Trichodesmium* filaments and 4 single *Hemiaulus*-*Richelia* symbiotic cells. Thus our observations are in agreement with earlier studies showing that most filaments are collected by their length rather than their diameter during sampling[34].

Several new locations to our knowledge not previously reported in diazotroph databases were identified (see below; Fig. 3, Supplementary Figs. S3 and S4). In addition, we detected colonies of *Crocosphaera*-like cells and others living in the more rarely reported diatom *Climacodium*[12–14]. Notably, a few *Crocosphaera* cells (1–2 cells) were detected by their strong phycoerythrin signal emitted in the alexa 546 channel while both the *Crocosphaera* cells and chloroplast of each *Climacodium* host emitted autofluorescence signals in the red channel (638 nm), demonstrating the combination of signals is a key attribute in detection of inconspicuous plankton by the image recognition model (Fig. 1).

Ranges in abundances based on image analyses for the 3 main DDAs: *Hemiaulus*-*Richelia*, *Rhizosolenia*-*Richelia*, and *Chaetoceros*-*Calothrix*, were low and are representative of background densities (e.g., 1.5–20 symbiotic cells $L^{-1}$) (Fig. 3 and Supplementary Fig. S3a). The low densities and detection, especially *Chaetoceros*-*Calothrix* which can form long chains (>50 cells chain$^{-1}$) and the larger *Rhizosolenia*-*Richelia* symbioses, were not surprising given the 180 μm sieve used in the

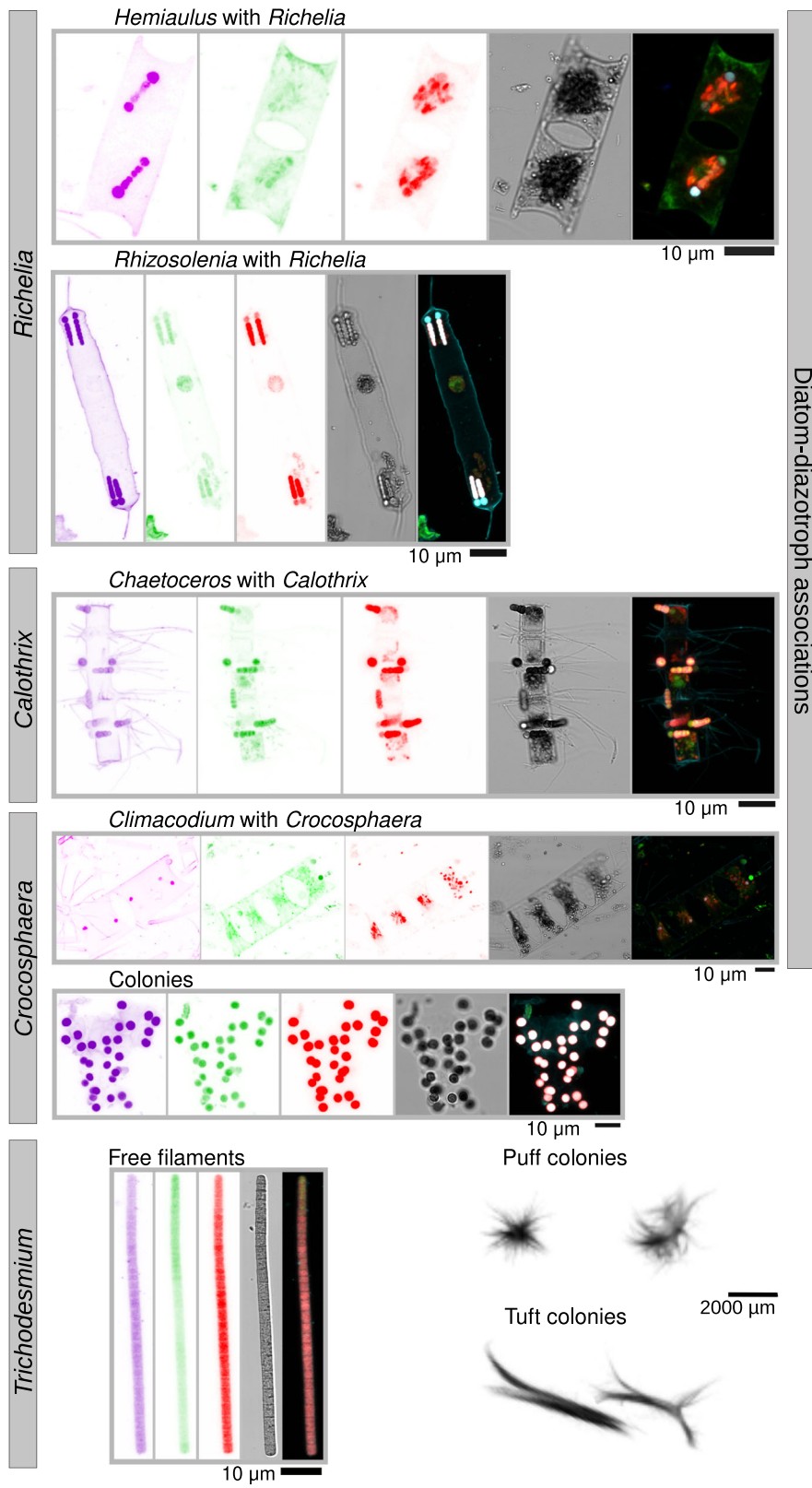

sample processing. Although *Hemiaulus-Richelia* was the most frequently detected, its chains were often short (1–2 cells), and sometimes cell integrity was compromised (i.e., broken frustules; poor autofluorescence). Variation in the number and length of the symbiont filaments (trichomes) was also observed and was dependent on which host the *Richelia/Calothrix* was associated (Fig. 2), consistent with previous observations[13]. Free *Richelia*

and *Calothrix* filaments were also found (Supplementary Fig. S5), which are less often reported in the literature[35], but are not unexpected for facultative symbionts[36,37].

DDAs and free *Richelia/Calothrix* filaments were broadly distributed and detected in several new locations[19,20] to our knowledge, including the Indian Ocean (IO), western South Atlantic Ocean (SAO), the South Pacific gyre, and the Pacific side

**Fig. 1 Imaging observations of diazotrophs in *Tara* Oceans samples.** Images were obtained by environmental high content fluorescence microscopy (eHCFM; ref. [30]), with the exception of *Trichodesmium* colonies, which were detected in situ using an Underwater Vision Profiler 5 (UVP5; ref. [33]). From left to right, the displayed channels for each micrograph correspond to: cell surface (cyan, AlexaFluor 546), cellular membranes (green, DiOC6), chlorophyll autofluorescence (red), the bright field, and the merged channels. The displayed *Hemiaulus-Richelia* association was detected at station TARA_80 in the South Atlantic Ocean, *Rhizosolenia-Richelia* at TARA_53 in the Indian Ocean, *Chaetoceros-Calothrix* at TARA_131 (ALOHA) in the North Pacific Ocean, *Climacodium-Croscophaera* at TARA_140 in the North Pacific Ocean, the *Croscophaera*-like colony at TARA_53 in the Indian Ocean, the *Trichodesmium* filament at TARA_42 in the Indian Ocean, and the *Trichodesmium* colonies at TARA_141 and TARA_142 in the North Atlantic Ocean. Each micrograph is representative of the following number of images obtained in the current study: 350 *Hemiaulus-Richelia*, 56 *Rhizosolenia-Richelia*, 8 *Chaetoceros-Calothrix*, 1 *Climacodium-Crocosphaera*, 150 *Crocosphaera*-like colonies, 605 *Trichodesmium*-free filaments, 115 puff and 95 tuft colonies of *Trichodesmium*.

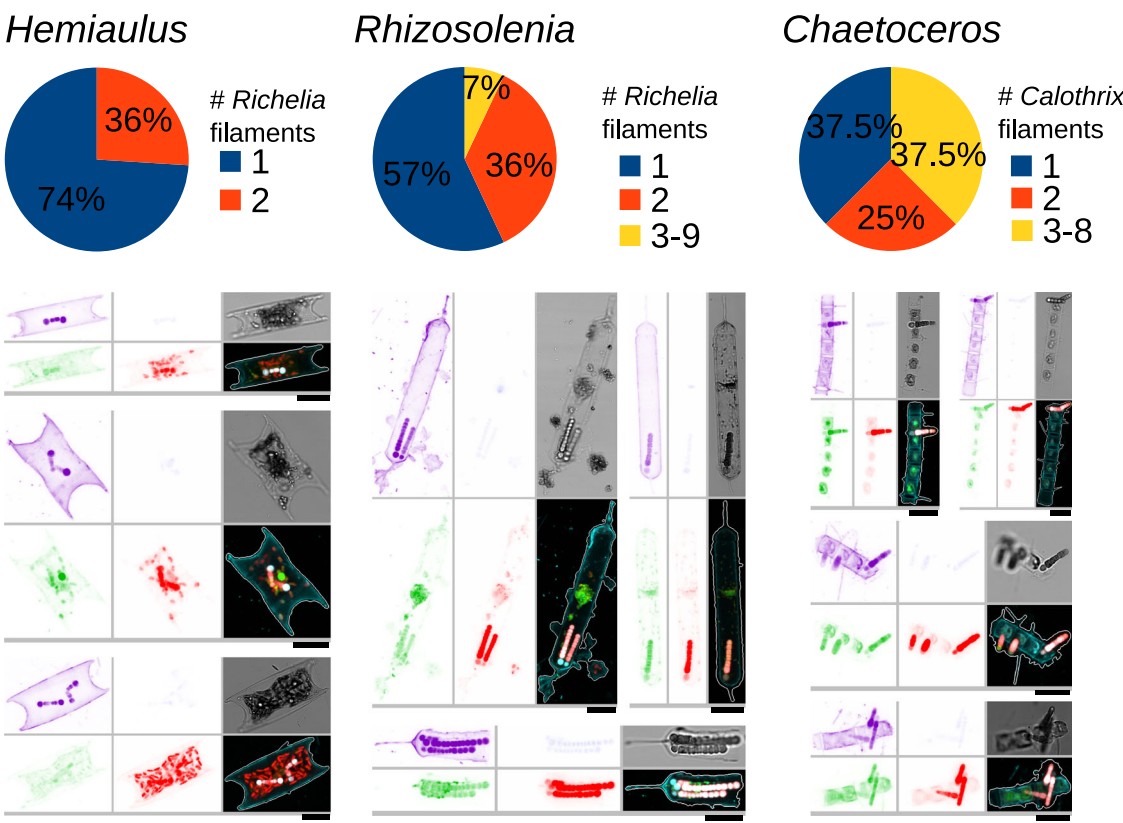

**Fig. 2 Variation in the number of *Richelia/Calothrix* filaments among the diatom-diazotroph associations observed by high-throughput confocal microscopy.** Examples of images are shown. Clockwise from top-left, the displayed channels for each micrograph correspond to: cell surface (cyan, AlexaFluor 546 dye), DNA (blue, Hoechst dye), the bright field, cellular membranes (green, DiOC6 dye), chlorophyll autofluorescence (red), and the merged channels. The size bar at the bottom left of each microscopy image corresponds to 10 µm.

of the Panama Canal (Supplementary Figs. S3 and S5). DDAs were concentrated in the surface, with the exception of two deeper samples of *Hemiaulus-Richelia*, with densities as high as in the surface. The two exceptions were the 108-m depth sample from the Hawaii Ocean Time Series station ALOHA (A Long-Term Oligotrophic Habitat Assessment; TARA_131; North Pacific Subtropical Gyre) and the 38-m depth one from TARA_143 (Gulf Stream, North Atlantic) (Fig. 3b). Seasonal blooms of DDAs are well known at station ALOHA, with observations of DDAs in moored sediment traps below the photic zone[38–40]. However, observations of symbiotic diatoms in the Gulf Stream are more rare[41].

We observed densities of 1–40 *Trichodesmium*-free filaments L$^{-1}$. Similar to the DDAs, long free filaments of *Trichodesmium* were likely undersampled due to the collection and processing (35% of them are >180 µm in length). Free filaments of *Trichodesmium* co-occurred with DDAs in most stations from the IO and North Pacific Ocean (NPO) (Fig. 3a

and Supplementary Fig. S4), and they were also observed at sites where DDAs were not detected, such as in the Pacific North Equatorial Current (TARA_136). Tens to hundreds of *Trichodesmium* filaments often aggregate into fusiform-shaped colonies usually referred to as 'tufts' or 'rafts' or round-shaped colonies called 'puffs' (from 200 µm to 5 mm) (Fig. 1). These dimensions were detectable and quantifiable by in situ imaging using the UVP5[33] (Fig. 3 and Supplementary Fig. S4a). A total of 220 images were clearly curated as tuft or puff colonies, with the major axis ranging from 1 to 10 mm and the minor axis from 0.4 to 4 mm (https://www.ebi.ac.uk/biostudies/studies/S-BSST529). Our stringent annotation probably underestimated *Trichodesmium* colony abundances (see below the correlation with metagenomes). Colony densities were more prevalent in the NAO and NPO, while free filament densities were more abundant in the IO (Fig. 3a and Supplementary Fig. S4a), probably related to the enhanced colony formation of *Trichodesmium* under nutrient limitation (Supplementary

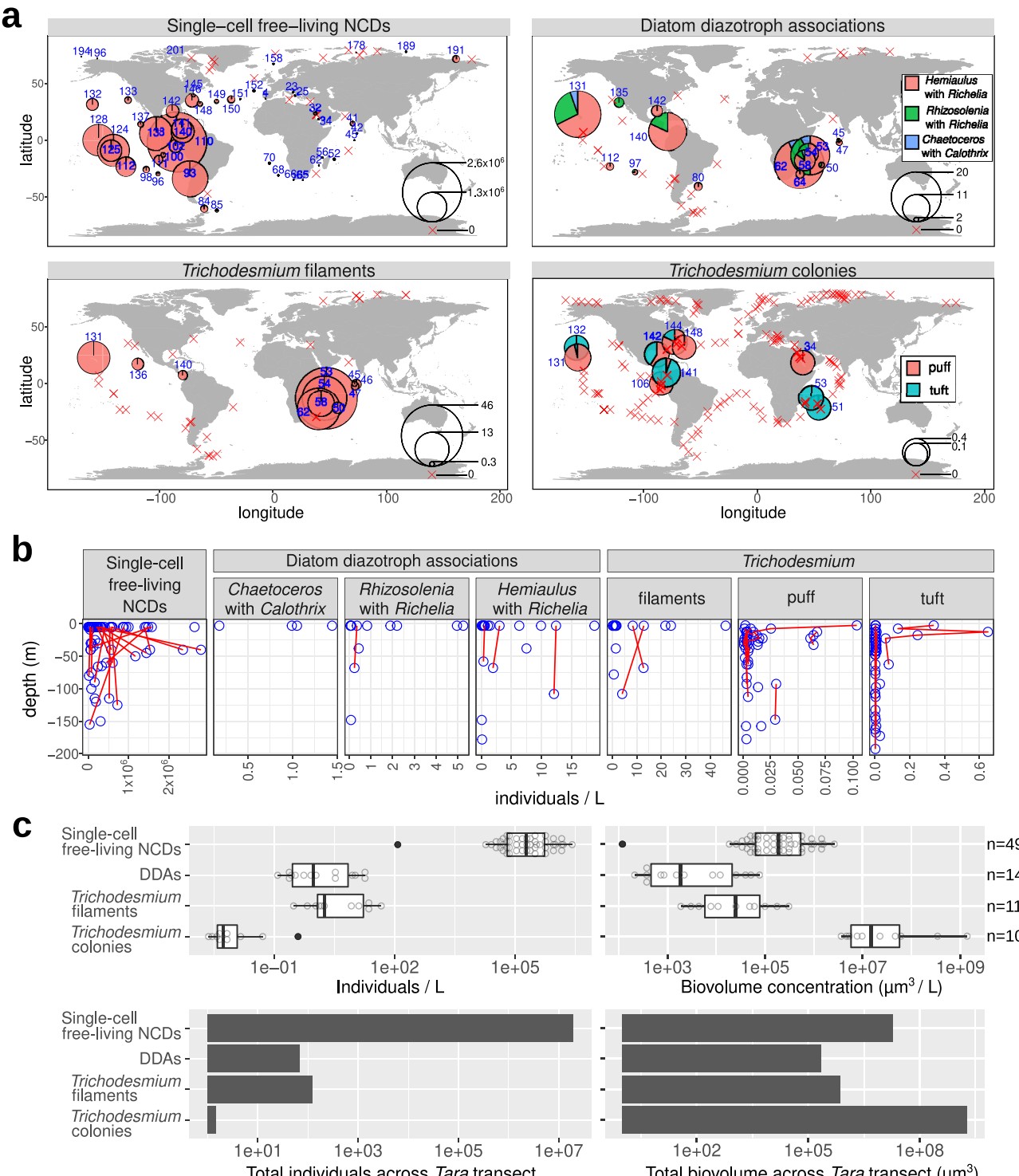

**Fig. 3 Abundance and distribution of diazotrophs by quantitative imaging methods. a** Biogeography in surface waters. Bubble size varies according to the corresponding diazotroph concentration (individuals/L), while red crosses indicate their absence. Station labels with detection of diazotrophs are indicated in blue. **b** Depth partition. Samples from the same geographical site are connected by red lines. **c** Distribution of individual abundances and biomass in surface waters. Single-cell free-living non-cyanobacterial diazotrophs (NCDs) were quantified by merging flow cytometry counts with *nifH/recA* ratio from metagenomes from size fraction 0.22–1.6/3 μm and assuming an average cellular biovolume of 1 μm³ based on the cell dimensions reported in the literature for cultured NCDs[102–107]. The detection and biovolume determinations of diatom-diazotroph associations (DDAs) and *Trichodesmium*-free filaments were carried out by high-throughput confocal microscopy in samples from the 20–180 μm size fraction. In the case of *Trichodesmium* colonies, it was determined using in situ images from the UVP5. Boxplots depict the 25–75% quantile range of the dataset without zeros (the corresponding biologically independent seawater samples are indicated in the plot), with the center line depicting the median (50% quantile); whiskers encompass data points within 1.5× the interquartile range. Source data are provided as a Source data file.

Fig. S6), and confirms the tendencies observed in culture experiments[42].

Single-cell free-living NCDs were estimated by combining flow cytometry estimates of free-living bacterial densities with diazotroph relative abundances derived from metagenomic sequencing of the 0.22–1.6/3 μm plankton size fractions (see "Methods"). We detected concentrations up to ~$2.8 \times 10^6$ cells L$^{-1}$, with the highest values in the Pacific Ocean (Fig. 3a). Our estimates agree with recent reports based on the reconstruction of metagenome-assembled genomes[16].

Combining the imaging datasets from *Tara* Oceans (flow cytometry, confocal microscopy, and UVP5) enabled the conversion of abundances to estimates of biovolumes for the NCDs, *Trichodesmium*, and DDAs (Fig. 3c). Single-cell free-living NCDs are by far the most abundant diazotrophs in the surface ocean, however, *Trichodesmium* dominates in terms of biovolume by two orders of magnitude over NCDs (Fig. 3c). Cell density and biovolume of NCDs have not been previously reported at a global scale, thus the work presented here expands our understanding on the relative contributions for these recently recognized important diazotrophs.

**Diazotroph diversity and abundance using metagenomes from size-fractionated plankton samples.** To gain further insights into the abundance and distribution of diazotrophs across the whole plankton size spectrum, we compared the imaging data with metagenomic reads from the 5 main size fractions mapped against a comprehensive catalog of 29,609 unique *nifH* sequences (see "Methods"). The *nifH* catalog represents most of the genetic diversity reported for diazotroph isolates and environmental clone libraries (although it has some redundancy; see "Methods"), with 30% of the sequences derived from marine environments and the rest from terrestrial and freshwater habitats. Around 2.5% of these *nifH* sequences (762 out of 29,609) mapped with at least 80% similarity to the 1164 metagenomes, retrieving a total of 96,215 mapped reads. Of the 762 sequences, 167 retrieved only one read. Mapped *nifH* reads were detected in slightly more than half of the samples (66% or 771 of 1164 metagenomes), which highlights the broad distribution of diazotrophs in the *Tara* Oceans datasets (blue circles in Fig. 4a for surface waters; Supplementary Data 1).

The read abundance ratio of *nifH*/*recA* was used to estimate the relative contribution of diazotrophs (see "Methods"). Our analysis shows both a dramatic increase (up to 4 orders of magnitude) in diazotroph abundance and a compositional shift towards the larger size classes of plankton (Fig. 5). For example, diazotrophs comprise only a small proportion of the bacterial community in the 0.22–1.6/3 μm size fraction (0.004–0.8%), however, they increase to 0.003–40% in the 180–2000 μm size range (Fig. 5a). The increase is coincident with a change in taxonomy (Fig. 5b, c, Supplementary Data 1): proteobacteria and planctomycetes are the main components in the 0.22–1.6/3 μm size fraction (0.004–0.08% and 0.005–0.4%, respectively), while cyanobacterial diazotrophs dominate in the larger size fractions, including both filamentous (*Trichodesmium* and others) and non-filamentous types (free-living and symbiotic) (0.2–45% and 0.2–2%, respectively). A remarkable congruence was observed between the quantification based on imaging and metagenomic methods for the larger cyanobacterial diazotrophs (Fig. 6, Supplementary Figs. S3ab and S4ab). Hence, a fully reversible pipeline from images to genomics and genomics to images that allows each to inform the other was developed. The image analysis enables one to quickly identify which parallel metagenomic (or metatranscriptomic) sample(s) should contain a particular diazotroph. For populations like the cyanobacterial diazotrophs which are comparatively less abundant, this approach will reduce search time in genetic analyses.

The majority (95%) of the total recruited reads mapping to the *nifH* database corresponded to 20 taxonomic groups: 5 cyanobacteria, 2 planctomycetes, and 13 proteobacteria. For the NCDs, the 2 planctomycetes and 7 of the 13 proteobacterial types corresponded to recent metagenome-assembled genomes (named HBD01 to HBD09[16]) which additionally were among the top contributors to the *nifH* transcript pool in the 0.22–1.6/3 μm size fraction of *Tara* Oceans metatranscriptomes[22]. We also found these taxa in the larger size fractions (Figs. 5c and 7). The 0.8 μm pore-size filter enriches for larger bacterial cells, while letting pass smaller diameter cells (including the more abundant taxa: SAR11 and *Prochlorococcus*). However, it is interesting that NCDs were detected in the three largest size fractions (5–20, 20–180, or 180–2000 μm), suggesting NCDs attach to particles (e.g., marine snow, fecal pellets)[43] and/or larger eukaryotic cells/organisms, aggregate into colonies[44], or are possibly grazed by protists, suspension feeders and/or copepods[45].

The main cyanobacterial taxa corresponded to *Trichodesmium*, *Richelia*/*Calothrix*, and UCYNs (UCYN-A1, UCYN-A2, and *Crocosphaera*). *Trichodesmium* represented the highest number of reads for *nifH* among all diazotrophs and constituted up to 40% of the bacterial community in the three largest size fractions (Figs. 5c and 7). *Richelia nifH* reads were also detected mainly in 5–20, 20–180, and 180–2000 μm, while *Calothrix* was limited to 20–180 μm. Both observations are consistent with the expected association of *Richelia*/*Calothrix* with large and small diatoms (*Hemiaulus*, *Rhizosolenia*, *Chaetoceros*) (Figs. 1, 2, 5c, and 7), and the occasional report of free filaments. Free filaments were also detected in the confocal analyses (Supplementary Fig. S5; https://www.ebi.ac.uk/biostudies/studies/S-BSST529). Relative abundance of UCYN-A1 was highest in the smaller size fractions 0.2–1.6/3 μm and 0.8–5 μm, in accordance with the expected host cell size (1–3 μm)[11,46,47]. However, UCYN-A1 was also detected in the larger size fractions (5–20, 20–180, and 180–2000 μm) (Figs. 5c and 7) and suggests UCYN-A was grazed[45] and/or associated with larger particles (e.g., marine snow or aggregates[48]), which may subsequently sink to the deep ocean (see next section). *Crocosphaera* was also found in multiple size fractions (0.8–5, 5–20, 20–180, and 180–2000 μm), which was expected given its diverse lifestyles: free-living, colonial, and symbiotic with large *Climacodium* diatoms (Fig. 1)[12–14].

Unexpectedly, *nifH* sequences with high similarity to 'spheroid bodies' (Supplementary Fig. S7a) of a few freshwater rhopalodiacean diatoms[49–51], were recruited in the surface waters of the 20–180 μm size fraction (Figs. 5c and 7). These observations were noted in multiple ocean basins (IO, SPO, and SAO; Supplementary Fig. S7b). Genome analysis of 'spheroid bodies' indicates the loss of photosynthesis[52] and other metabolic pathways common to free-living cyanobacteria, and hence they are deemed obligate to their respective hosts. Therefore, we would not expect them in the smaller size fractions. To our knowledge, this is the first genetic evidence for the presence of these populations in marine waters. Detection levels were however low (~0.5% of the total bacterioplankton community) (Figs. 5c and 7), and consistent with the expected diatom host cell diameters (~30–40 μm[53]). Using published images of symbiotic rhopalodiacean diatoms[54], we manually selected similar images in the samples of interest and used them as a training set. The image recognition model predicted several images of pennate diatoms containing round granules without chlorophyll autofluorescence (Supplementary Fig. S7c). However, our predictions are preliminary, and require further validation by other methods that are outside the scope of this work to verify whether they are indeed diazotrophic symbionts.

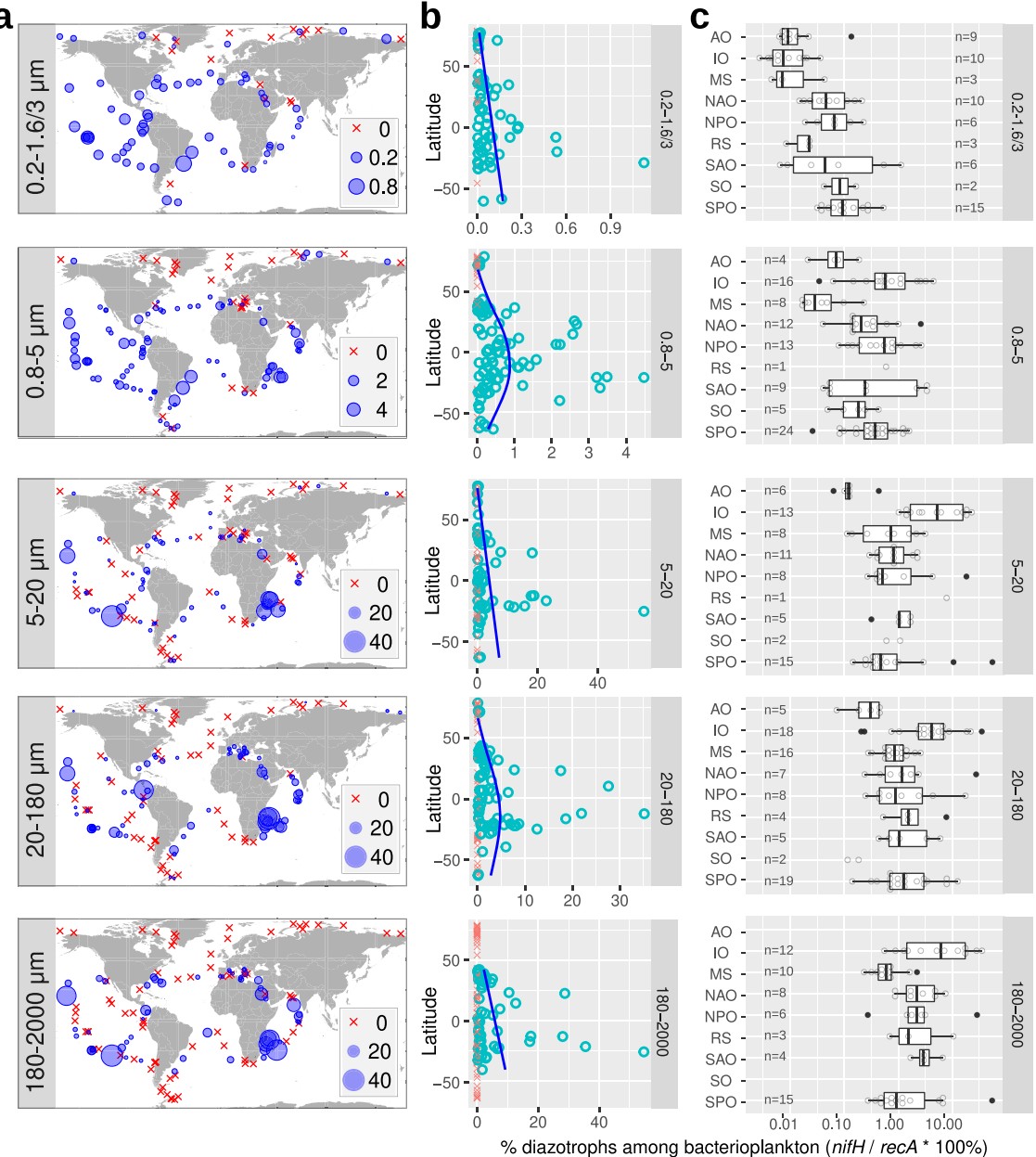

**Fig. 4 Biogeography of diazotrophs in surface waters using metagenomes obtained from different size-fractionated samples.** The percentage of diazotrophs in the bacterioplankton community was estimated by the ratio of metagenomic read abundance between the marker genes *nifH* and *recA*. **a** Biogeography. The bubble size varies according to the percentage of diazotrophs, while red crosses indicate absence (i.e., no detection of *nifH* reads). **b** Latitudinal abundance gradient. The blue lines correspond to generalized additive model smoothings. **c** Ocean distribution. MS Mediterranean Sea, IO Indian Ocean, SAO South Atlantic Ocean, SO Southern Ocean, SPO South Pacific Ocean, NPO North Pacific Ocean, NAO North Atlantic Ocean, AO Arctic Ocean. Boxplots depict the 25–75% quantile range of the dataset without zeros (the corresponding biologically independent seawater samples are indicated in the plot), with the center line depicting the median (50% quantile); whiskers encompass data points within 1.5× the interquartile range. Source data are provided as a Source data file.

**Evidence of polyploidy**. To date, polyploidy has been reported in NCDs of soil[55,56], in some heterocystous and UCYN cyanobacteria[56–59], and in a field/cultured study of *Trichodesmium*[60]. It, however, remains unknown in marine NCDs, diazotrophic UCYNs, and *Richelia/Calothrix* populations. Hence we cannot discount that the *nifH* abundances for the diazotrophs was not influenced by ploidy. Here we have attempted to estimate whether ploidy influences *Trichodesmium* and *Richelia/Calothrix* abundances since we have quantifications for both based on imaging and genetic methods from the same samples (Fig. 6). However, it is important to note that the

metagenomic sampling from *Tara* Oceans was not specifically designed to quantify metagenomic signals per seawater volume due to the lack of 'spike-ins' or DNA internal standards, and thus we have made assumptions in order to approximate estimates using the reported sampled seawater volumes and the quantity of extracted DNA (see "Methods"). Based on these assumptions, our results show the polyploidy for *Trichodesmium* is 5 (median of 15 points) and 2 for *Richelia/Calothrix* (median of 21 points), but both taxa show variability of polyploidy values according to the sample (range of 0.3–58; Supplementary Data 2). These results are within the range of polyploidy reported previously for field

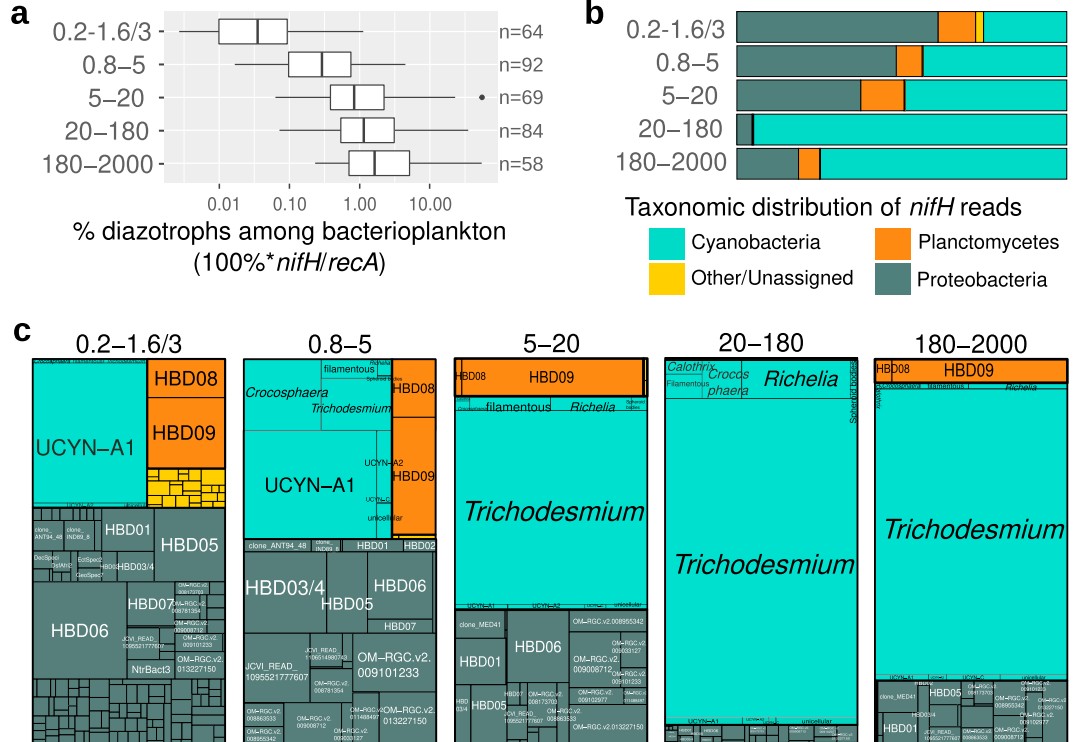

**Fig. 5 Abundance of diazotrophs in surface waters using metagenomes obtained from different size-fractionated samples. a** Diazotroph abundance. The percentage of diazotrophs in the bacterioplankton community was estimated by the ratio of metagenomic read abundance between the marker genes *nifH* and *recA*. **b** Taxonomic distribution of the *nifH* reads. **c** Taxonomic distribution at deeper resolution. HBD01 to HBD09 (heterotrophic bacterial diazotrophs) corresponds to the metagenome-assembled genomes from <3-µm size fractions[16]. Boxplots depict the 25–75% quantile range of the dataset without zeros (the corresponding biologically independent seawater samples are indicated in the plot), with the center line depicting the median (50% quantile); whiskers encompass data points within 1.5× the interquartile range. Source data are provided as a Source data file.

populations of *Trichodesmium* (range of 1–120; ref. [60]), but below those observed in culture conditions (range 639–697; ref. [60]). When comparing the polyploidy of *Trichodesmium* vs *Richelia*/*Calothrix* in the 12 samples in which both are detected, the estimate for *Trichodesmium* is on average 4 times the estimate for *Richelia*/*Calothrix* (range 0.1–15) (Supplementary Data 2). Therefore, *Trichodesmium* dominance over *Richelia*/*Calothrix* is overestimated in the metagenomes due to the higher polyploidy levels of the former. Equivalent values for NCDs and UCYNs will be needed to evaluate the effect of polyploidy in the whole diazotroph community. In addition, it is important to note in previous works suggest that the degree of ploidy depends on the growth conditions, nutrient status, developmental stage, and cell cycle[55–60].

**Insights into environmental distribution and depth partitioning of diazotrophs.** Diazotroph abundance was latitudinally influenced and shows expected higher relative abundances in tropical and subtropical regions, and a decrease at the equator where upwelling and higher dissolved nutrients are expected (Fig. 4). This pattern is congruent with decades of field observations (e.g., NAO, NPO) as well as modeling efforts[20,61,62], and the correlation analyses of environmental and physico-chemical variables measured during *Tara* Oceans (Fig. 8). Temperature and nutrient availability are common factors which govern diazotroph abundances[8,20,63]. Iron is also expected to be important due to the high iron requirement of the nitrogenase enzyme[64,65], therefore it was unexpected to find a less robust relationship between diazotroph abundances and modeled dissolved iron concentrations (Fig. 8a). However, we cannot discount

inaccuracies of modeled values relative to in situ bioavailable Fe concentrations.

We further analyzed abundance and distribution patterns in the deeper depth samples (0–200 m and 200–1000 m, respectively). The higher numbers of $N_2$-fixing cyanobacteria detected in the surface (5 m) compared to the deep chlorophyll maximum (DCM; 17–188 m) in both the metagenomic and imaging datasets confirms expected distributions (Figs. 3b and 8c, also compare Fig. 9 and Supplementary Fig. S8). However, detection of both *Trichodesmium* and the DDA symbionts were nonetheless significant in some DCM samples from diverse regions: IO, SPO, and Red Sea (RS) (Supplementary Fig. S8). DDAs are expected at depth given the reported rapid sinking rates, and observations in moored sediment traps (station ALOHA: refs. [39,40,66]). Traditionally *Trichodesmium* has been considered to have a poor export capacity[67], however, recently there are contrary reports, and therefore *Trichodesmium* is also expected at depth[40,68,69]. Increased abundances of *Crocosphaera* co-occurred in the DCM of IO samples in the larger (5–20 µm) size fraction; indicative of the colonial and/or symbiotic lifestyles previously reported for *Crocosphaera* (Fig. 1; refs. [12,70]). Unlike the phototrophic diazotrophs, the distribution of NCDs had no apparent depth partitioning in the epipelagic layer (Fig. 8c).

A relatively high number of *nifH* reads was detected in mesopelagic samples (110 out of 128—or 86%—of mesopelagic samples, Supplementary Fig. S9). Although BNF and *nifH* expression has been previously reported at depth, most measurements have been made in oxygen minimum zones (OMZs; where low-oxygen waters are found) and oxygen-deficient zones (ODZs; where oxygen concentrations are low enough to induce anaerobic metabolisms)[17,22]. Here, *nifH* sequences mapped from many

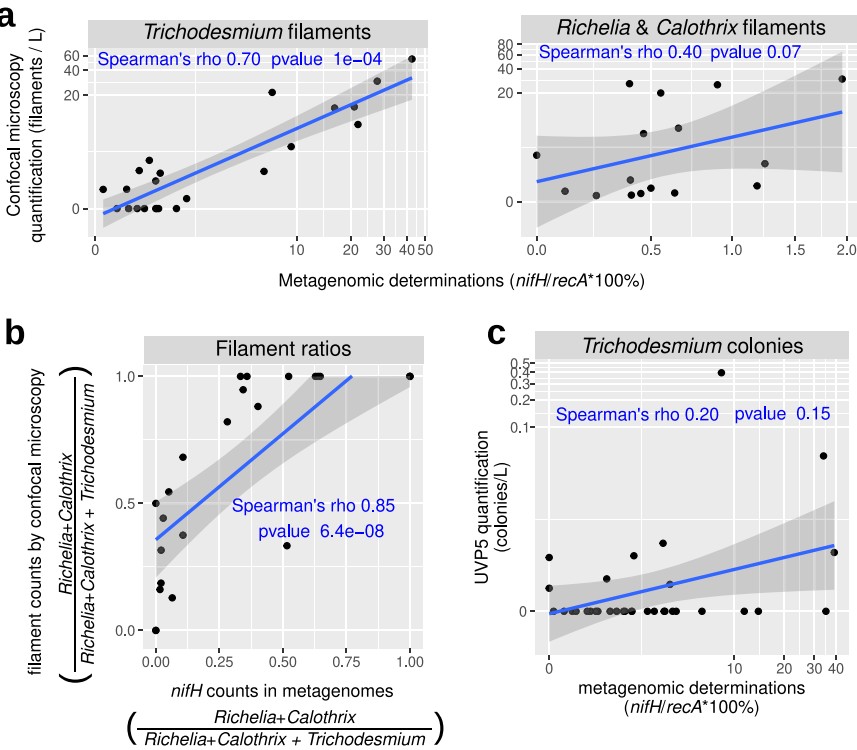

**Fig. 6 Correlation analysis between diazotroph quantifications by imaging and molecular methods. a, b** Comparison between environmental high content fluorescence microscopy (eHCFM[30]) and metagenomics. *Calothrix*, *Richelia*, and *Trichodesmium* in samples from size fraction 20–180 µm were measured by quantification of high-throughput confocal microscopy images (filaments L$^{-1}$) and by metagenomic counts (% of diazotrophs in the bacterioplankton community by the ratio between the marker genes *nifH* and *recA*). **a** Correlation of relative abundances in metagenomes and absolute abundances by confocal microscopy for the three taxa. **b** Correlation between the ratio of abundances between taxa. **c** Comparison between Underwater Vision Profiler 5 (UVP5[33]) and metagenomics. *Trichodesmium* colonies were measured by UVP5 quantification (colonies L$^{-1}$) and by metagenomic counts in the 180–2000 µm size-fractionated samples. Spearman's rho correlation coefficients and uncorrected one-side p-values are displayed in blue. Source data are provided as a Source data file.

mesopelagic samples outside of OMZs and ODZs including in SPO, NPO, NAO, and SAO (Supplementary Fig. S9). The majority of *nifH* sequences correspond to proteobacteria, however, diazotrophic cyanobacteria were also detected (Supplementary Fig. S9). In particular, 44% of total *nifH* reads in mesopelagic samples at TARA_78 and 6% at TARA_76 (of 0.2–3 µm size fraction) in SAO corresponded exclusively to UCYN-A (Supplementary Fig. S9, Supplementary Data 1). The maximum numbers of UCYN-A reads in all *Tara* Oceans samples were detected in the corresponding surface samples at these stations (Fig. 9; see below), suggesting a surface bloom. Most reports indicate the presence and activity of UCYN-A in the sunlit zone, with the exception of a study reporting UCYN-A *nifH* sequences in shallow water sediments of the north east Atlantic ocean (seafloor 38–76-m depth)[71]. Our observation of UCYN-A at 800-m depth in the open ocean suggests that this symbiosis could contribute to carbon export despite its small cell diameter, or that other physical processes (e.g., subduction, downwelling, upwelling) in this ocean region were influencing the high detection of UCYN-A at depth.

**Global ocean biogeography of diazotrophs.** Several regions displayed high densities or "hotspots" of diazotrophs which have not been previously sampled (Figs. 4 and 9, Supplementary Fig. S10). For example, in the Mozambique Channel between Madagascar and the African continent, diazotrophs constituted up to 30–40% of the bacterioplankton in the larger size fraction samples (TARA_50 to TARA_62; Figs. 4 and 9). The latter molecular results were confirmed by images

of both *Trichodesmium* and symbiotic diazotrophs (Fig. 3a). Another example is the SAO near South America (TARA_76, TARA_78, and TARA_80), where UCYN-A reached 3–4% of the bacterioplankton population in the 0.8–5 µm size fraction (Figs. 4 and 9). These zones from IO and SAO represent previously undersampled regions for diazotrophs (Supplementary Fig. S1), which also lack quantitative rate measurements for BNF[19].

The highest abundance of free-living single-cell NCDs (0.2–3 µm) corresponded to ~0.5% of the bacterioplankton in the wake of the Marquesas archipelago in the equatorial PO (TARA_123; Fig. 9). This station was reported recently as a surface planktonic bloom triggered by natural iron fertilization[72]. Other high-density areas corresponded to a few stations in the SPO (surface samples of TARA_98 and TARA_99 and DCM of TARA_102), where high abundances of proteobacteria and planctomycetes (4–33% and 8–9%, respectively) were found in larger size fractions (Fig. 9 and Supplementary Fig. S8), which likely results from association of NCDs to sinking particles. Moreover, TARA_102 is located in the Peruvian upwelling area, a region previously reported for NCDs and/or BNF activity associated with the OMZ[73–75]. These results are congruent with recent reports from the subtropical Pacific of highly diverse NCDs, some associated with sinking particles[18,75–77]. We can therefore expand the distribution of potentially particle-associated NCDs to several other ocean basins (NAO, AO, IO). Our findings emphasize the dominance and persistence of NCDs in larger size fractions of both surface and DCM, which is new information to our knowledge and warrants further investigation.

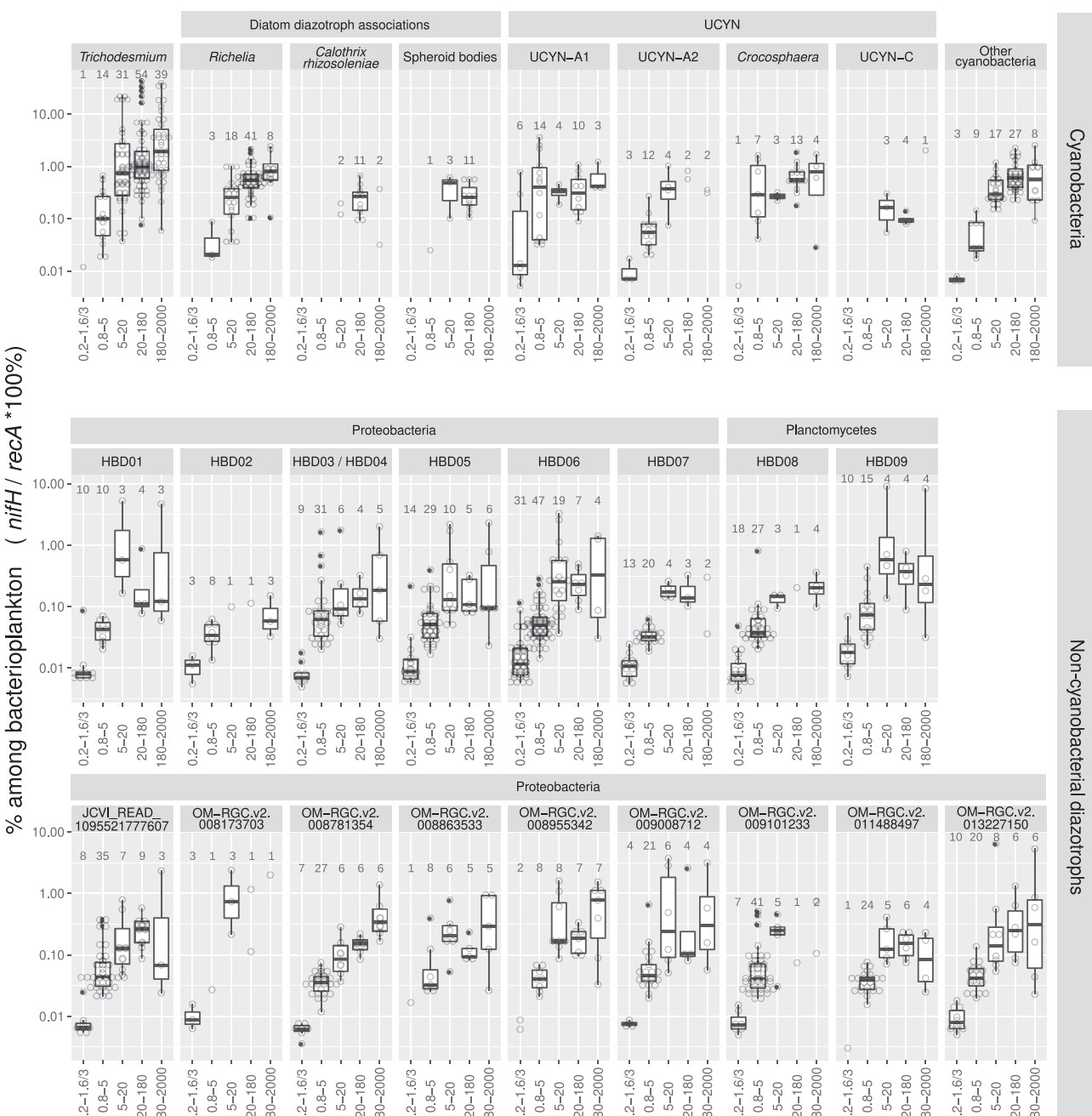

**Fig. 7 Distribution of the main diazotroph taxa across metagenomes obtained in different size-fractionated samples from surface waters.** For each taxon, the percentage in the bacterioplankton community is estimated by the ratio of metagenomic read abundance between the marker genes *nifH* and *recA*. The lineages grouped into 'Other cyanobacteria' are displayed in the Source data File. The 'OM-RGC.v2' prefix indicates the *nifH* sequences assembled from the metagenomes of <3-μm size fractions[22], while HBD01 to HBD09 (heterotrophic bacterial diazotrophs) corresponds to the metagenome-assembled genomes from the same samples[16]. Boxplots depict the 25–75% quantile range of the dataset without zeros (the corresponding biologically independent seawater samples are indicated in the plot), with the center line depicting the median (50% quantile); whiskers encompass data points within 1.5× the interquartile range. Source data are provided as a Source data file.

Many regions contain a low abundance of diazotrophs. For example, the percentages of diazotrophs in the AO, the Southern Ocean (SO), and the Mediterranean Sea (MS) reached maximum values of only 0.4, 1, and 4%, respectively (Figs. 4c and 9). The highest diazotroph abundance in the AO corresponded to NCDs found in shallow waters (20–25 m depth) of the East Siberian Sea (TARA_191; Figs. 4c and 9), a biologically undersampled region. Combined, the results concerning distribution reported here emphasize the patchy biogeographical patterns of diazotrophs.

**Cyanobacterial diazotrophs are mainly found as assemblies of abundant groups.** With the exception of a few stations in IO, RS, NPO, and NAO where *Trichodesmium* was the main component of the mapped reads (Figs. 8b and 9, Supplementary Fig. S11), there was a general and consistent trend of co-occurrence for several cyanobacterial diazotrophs in multiple oceanic regions (Fig. 9 and Supplementary Fig. S11). For example, in the Red Sea (RS), southern IO, and at station ALOHA, both larger (*Trichodesmium*, *Richelia*) and smaller (UCYN-A, *Crocosphaera*) diameter diazotrophs were

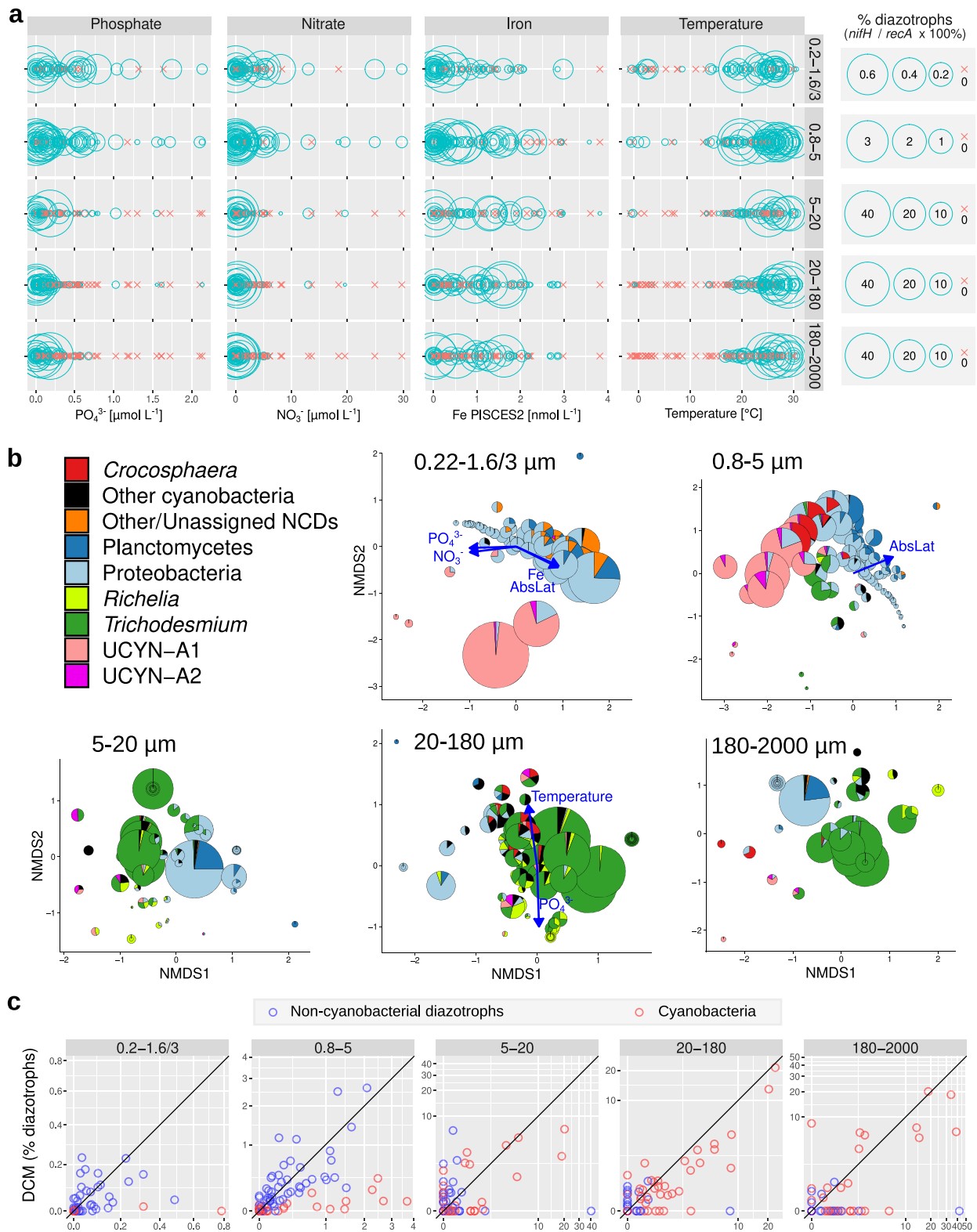

**Fig. 8 Environmental parameters and diazotroph distributions. a** Distribution across gradients of nutrients and temperature in surface waters. Blue circles correspond to samples with diazotrophs, while red crosses indicate absence (i.e., no detection of *nifH* reads). **b** NMDS analysis of stations according to Bray–Curtis distance between diazotroph communities of size-fractionated surface samples. Fitted statistically significant physico-chemical parameters are displayed (adjusted *P* value < 0.05). NMDS stress values: 0.07276045, 0.1122258, 0.1452893, 0.09693721, and 0.07969211. **c** Depth distribution. The scatter plots compare the diazotroph abundances between surface (5 m) and deep chlorophyll maximum (DCM; 17–180 m) for cyanobacteria (red points) and non-cyanobacterial diazotrophs (NCDs, blue points). Axes are in the same scale and the diagonal line corresponds to a 1:1 slope. Source data are provided as a Source data file.

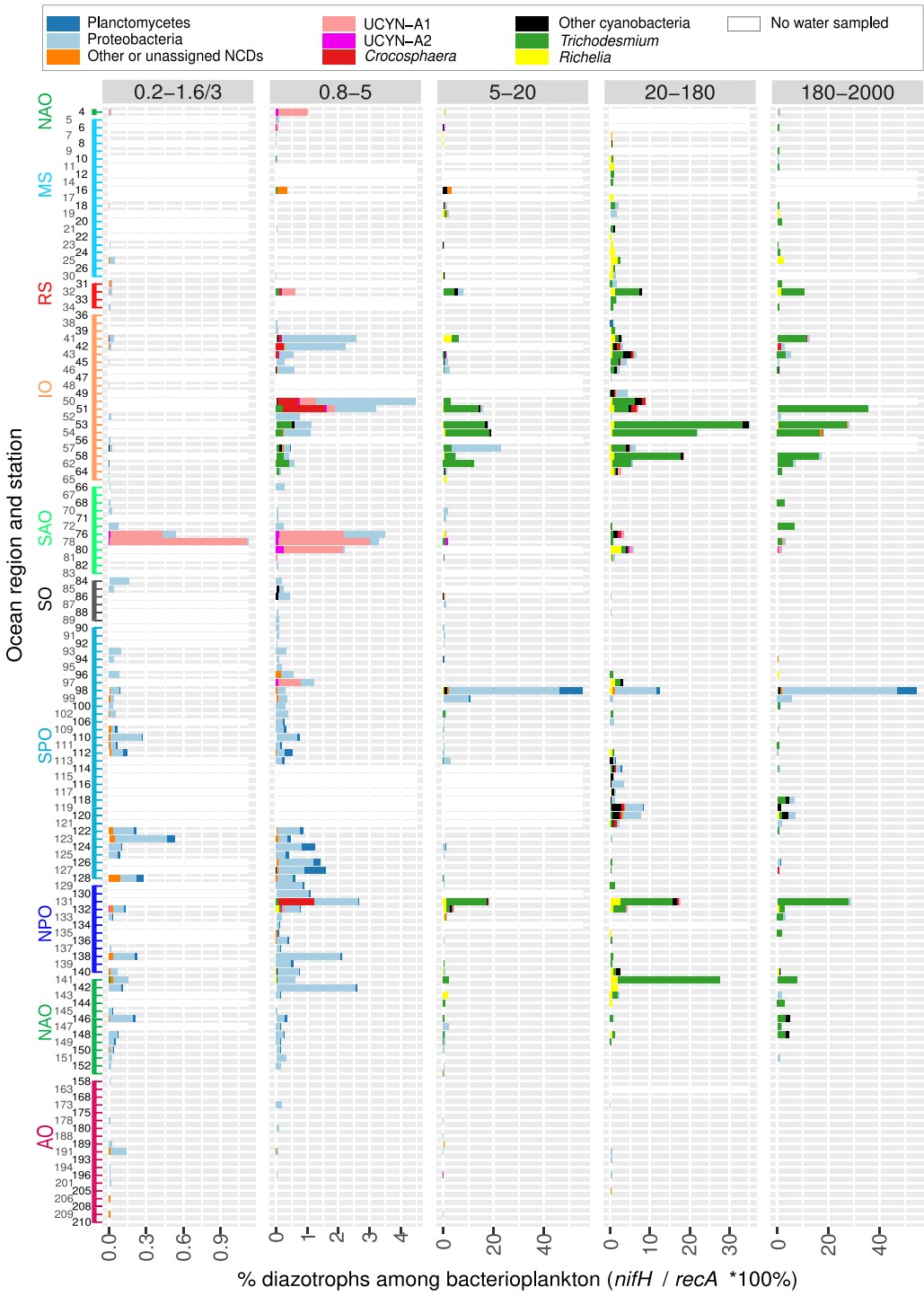

**Fig. 9 Diazotroph community based on metagenomes from size-fractionated surface samples.** The percentage of diazotrophs in the bacterioplankton community was estimated by the ratio of metagenomic read abundance between the marker genes *nifH* and *recA*. The bar color code shows the taxonomic annotation, while the absence of water sample is indicated by a white bar. The Y-axis shows the *Tara* Oceans stations and the ocean regions. MS Mediterranean Sea, IO Indian Ocean, SAO South Atlantic Ocean, SO Southern Ocean, SPO South Pacific Ocean, NPO North Pacific Ocean, NAO North Atlantic Ocean, AO Arctic Ocean. The equivalent figure showing the DCM water layer is shown in Fig. S2 (note the differences in scales between both figures, showing the higher relative abundance of diazotrophs in the surface layer). Source data are provided as a Source data file.

detected (Fig. 9 and Supplementary Fig. S11). In fact, UCYN-A has yet to be reported in the RS, while all other cyanobacterial diazotrophs have been reported previously, including surface blooms of *Trichodesmium* spp.[78–81]. On the contrary, only small diameter diazotrophs co-dominated in the SAO; for example, UCYN-A1 and UCYN-A2 were very abundant at stations TARA_76, TARA_78, and TARA_80 (Fig. 9 and Supplementary Fig. S11). The numerous and consistent observations of mixed diazotrophic assemblages of different sizes and lifestyles (colonial, free-living, symbiotic, particle-associated) highlight the need to consider how these traits enable co-occurrence.

**Ultrasmall diazotrophs consist of proteobacteria and are abundant in the Arctic Ocean**. Ultrasmall prokaryotes are unusual due to their reduced cell volume (these cells can pass through 0.22-μm filters, a size usually expected to exclude most microorganisms), and thus they are thought to have reduced genomes and to lack the proteins needed to carry out more complex metabolic processes. However, there is recent evidence that they do indeed participate in complex metabolisms[82]. To examine whether they may also contribute to marine BNF, we carried out the analysis of 127 metagenomes of <0.22 μm size-fractionated samples from different water layers.

A total of 79 *nifH* sequences in our database mapped with at least 80% similarity to these metagenomes, retrieving a total of 43,161 mapped reads, with the majority of the reads having high identity to proteobacterial *nifH* sequences. Of the 79 sequences, 15 retrieved only one read. Mapped *nifH* reads were detected in 92 of the 127 samples (72%), which highlights an unexpected broad distribution of ultrasmall diazotrophs (blue circles in Fig. 10a; Supplementary Data 1). Notably, when *nifH* reads were normalized by *recA* reads, we found that diazotrophs comprise up to 10% of the ultrasmall bacterioplankton, with the highest abundances detected in the AO, and in different water layers (Fig. 10a, b). This is remarkable considering that the lowest

diazotroph abundance in the other size fractions was detected in the AO (Figs. 4c and 9, Supplementary Figs. S8 and S9).

The majority (84%) of the total recruited reads mapping to the *nifH* database corresponded to two sequences assembled from the <0.22 μm size-fractionated metagenomes: OM-RGC.v2.008173703 and OM-RGC.v2.008955342. The former has 99% identity to *nifH* from the epsilon-proteobacterium *Arcobacter nitrofigilis* DSM7299[83] (hereafter *Arcobacter*), the second is more close in sequence identity to gamma-proteobacteria. The *Arcobacter*-like sequence only retrieved reads from surface and DCM, while the gamma-proteobacterium-like retrieved reads from surface, DCM, and mesopelagic (Fig. 10c). Both sequences also retrieved reads from other sizes fractions (Fig. 7 and Supplementary Fig. S12). Members of *Arcobacter* genus are either symbionts or pathogens[83], which is inconsistent with >9% of ultrasmall bacterioplankton associated with the <0.22 μm size fraction in the DCM waters of station TARA_158 (Supplementary Fig. S12). An additional proteobacterial sequence (OM-RGC.v2.008817394) almost exclusively retrieved reads from the <0.22 μm size-fractionated samples (Supplementary Fig. S12). Combined, these results prompt the inclusion of ultrasmall diazotrophs in future field surveys to quantify their potential ecological and biogeochemical importance.

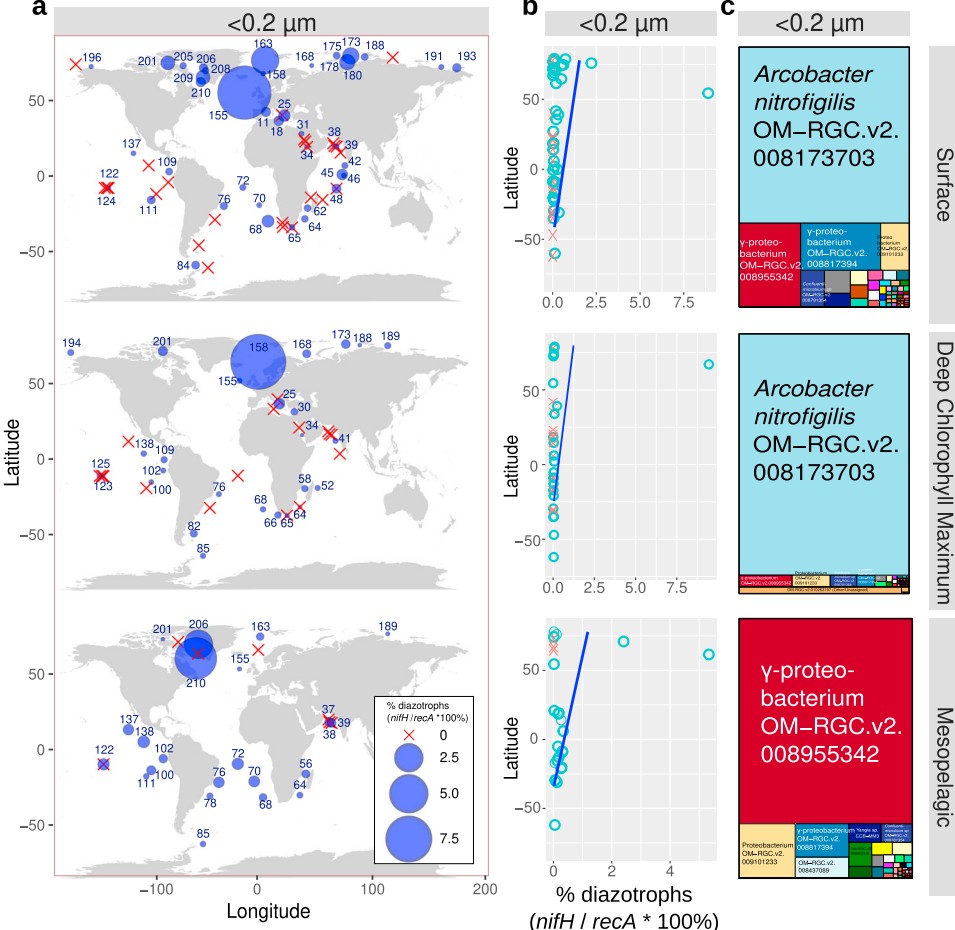

**Fig. 10 Detection of ultrasmall diazotrophs in metagenomes obtained from <0.22 μm size-fractionated samples of different water layers.** The percentage of diazotrophs among ultrasmall bacterioplankton was estimated by the ratio of metagenomic read abundance between the marker genes *nifH* and *recA*. **a** Biogeography. The bubble size varies according to the percentage of diazotrophs, while red crosses indicate absence (i.e., no detection of *nifH* reads). Station labels with diazotrophs detection are indicated in blue. **b** Latitudinal abundance gradient. Circles correspond to samples with diazotrophs, while crosses indicate absence. The blue lines correspond to generalized additive model smoothings. **c** Taxonomic distribution of the *nifH* reads. The 'OM-RGC.v2' prefix indicates the *nifH* sequences assembled from metagenomes of <3 μm size fractions[22], including <0.22 μm. Source data are provided as a Source data file.

In conclusion, the present work represents the first attempt, to our knowledge, to assess the diversity, abundance, distribution, and biovolume of diazotrophs at a global ocean scale using paired image and PCR-free molecular analyses. Unlike earlier studies, our work included the full biological and ecological complexity of diazotrophs: i.e., unicellular, colonial, particle-associated, symbiotic, cyanobacteria, and NCDs. Diazotrophs were found to be globally distributed and present in all size fractions, even among ultrasmall bacterioplankton (<0.22 μm), which were especially abundant in the AO. Unexpectedly, we detected sequences similar to obligate symbionts of freshwater diatoms nearly exclusively in the larger size fraction, in multiple and geographically distinct regions of the world's ocean. A major conclusion from our work is the identification of new hotspots for diazotrophs in previously undersampled regions. Historically and recently, the field has attempted to make paradigm shifts in favor of one dominant group (large vs. small diameter cyanobacterial diazotrophs). The data presented here on the contrary argues in favor of regions of co-existing assemblages of cyanobacterial diazotrophs with diverse lifestyles (colonial, free-living, symbiotic, particle-associated), while NCDs appear ubiquitously distributed. Overall, this work provides an updated composite of diazotroph biogeography in the global ocean, providing valuable information in the context of global change and the substantial anthropogenic perturbations to the marine nitrogen cycle.

## Methods

**Tara Oceans sampling.** *Tara* Oceans performed a worldwide sampling of plankton between 2009 and 2013 (Supplementary Fig. S1b). Three different water depths were sampled: surface (5-m depth), DCM (17–188 m), and mesopelagic (200–1000 m) (Supplementary Fig. S2). The plankton were separated into discrete size fractions using a serial filtration system[21]. Given the inverse logarithmic relationship between plankton size and abundance[21,84], higher seawater volumes were filtered for the larger size fractions (10–$10^5$ L; see Table 1 and Fig. 5 in ref. [21]). Taking into account that diazotrophs are less abundant than sympatric populations and have a wide size variation (Fig. 1), a comprehensive perspective requires analyses over a broad spectrum, which to date has been lacking. Five major organismal size fractions were collected: picoplankton (0.2–1.6 μm or 0.2–3 μm; named here 0.2–1.6/3 μm size fraction), piconanoplankton (0.8–5 μm or 0.8–2000 μm; named here 0.8–5 μm size fraction), nanoplankton (5–20 μm or 3–20 μm; named here 5–20 μm size fraction), microplankton (20–180 μm), and mesoplankton (180–2000 μm) (Supplementary Fig. S2)[21]. In addition, ultrasmall plankton (<0.22 μm) were also collected (Supplementary Fig. S2)[21]. The *Tara* Oceans datasets used in the present work are listed in Supplementary Fig. S2 and specific details about them and their analyses are described below.

**Read recruitment of marker genes in metagenomes.** The use of metagenomes avoids the biases linked to the PCR amplification steps of metabarcoding methods, and thus it is better for quantitative observations. This is especially important for protein-coding gene markers, such as *nifH*, which display high variability in the third position of most codons, and thus necessitate the use of highly degenerate primers for a broad taxonomic coverage[85]. The detection of low-abundance organisms, such as diazotrophs, is facilitated by the deep sequencing of the *Tara* Oceans samples (between ~$10^8$ and ~$10^9$ total metagenomic reads per sample)[22,24,25]. The 1326 metagenomes generated by the expedition are derived from 147 globally distributed stations and three different water layers: 745 metagenomes from surface, 382 from DCM (17–188 m), and 41 from the bottom of the mixed layer when no DCM was observed (25–140 m), and 158 from mesopelagic (200–1000 m) (Supplementary Fig. S2).

The read abundance of the single-copy gene *recA* was used to estimate the total bacterial community abundance in each sample (in contrast to the widely used 16S rRNA gene, which varies between 1 and 15 copies among bacterial genomes[86,87]). For simplicity, we assumed that *nifH* is also a single-copy gene, so the abundance ratio of *nifH*/*recA* provides an estimate for the relative contribution of diazotrophs to the total bacterial community. However, we realize that there are examples of 2–3 *nifH* copies in heterocyst-forming cyanobacteria such as *Anabaena variabilis* and *Fischerella* sp.[88,89], or in the firmicutes *Clostridium pasteurianum*[90], and that we are not taking into account the polyploidy effect.

The metagenomes were aligned against sequence catalogs of marker genes for diazotrophs (*nifH*) and bacteria (*recA*) (Supplementary Data 3 and 4). The analysis was carried out using bwa tool version 0.7.4[91] using the following parameters: -minReadSize 70 -identity 80 -alignment 80 -complexityPercent 75 -complexityNumber 30. The *nifH* sequence catalog (hereafter *nifH* database) was composed of 29,175 publicly available unique sequences from the laboratory of JP

Zehr (University of California, Santa Cruz, USA; version April 2014; https://www.jzehrlab.com). Although the Zehr database has some redundancy (9048 out of the 29,175 total unique sequences are retained when clustered at 95% identity using CDHIT-EST tool[92]), all non-identical sequences were used to maximize the number of metagenomic mapping reads. Moreover, this database was complemented with additional *nifH* genes retrieved from the sequenced genomes of Integrated Microbial Genome database (IMG)[93] and from different *Tara* Oceans datasets: Ocean Microbial Reference Gene Catalog version 2 (OM-RGC-v2[22]), assemblies[16] and 10 clones (this study). *nifH* clone libraries were built using DNA extracted from 0.2 to 1.6/3 μm size fraction plankton samples as described in Alberti et al.[24]. DNA samples from the same oceanic region were pooled together at a concentration of 10 ng/μl and used as template in a nested *nifH* PCR using degenerate primers (Supplementary Data 3 [94]). Five PCR products were purified and cloned using the TOPO-TA cloning kit (Invitrogen). Inserts were Sanger-sequenced bidirectionally using M13 Forward and M13 Reverse vector primers following manufacturer's recommendations. A total of 411 *nifH* clone sequences were generated, among which 148 showed less than 95% nucleotide identity with any previously published sequence. These 148 sequences were grouped into clusters sharing more than 99% nucleotide identity and represented by 10 consensus sequences (Supplementary Data 3). The *recA* sequences were obtained from sequenced genomes in IMG[93] and from OM-RGC-v2[22]. Homologous sequences were included in the two catalogs as outgroups to minimize false-positive read alignments. These latter sequences were retrieved from IMG, OM-RGC-v2, and the Marine Microbial Eukaryotic Transcriptome Sequencing Project (MMETSP[95]) using HMMer v3.2.1 with gathering threshold option (http://hmmer.org/). The outgroups for *recA* consisted of sequences coding for the RecA Pfam domain (PF00154), including RADA and RADB in Archaea, RAD51, and DCM1 in eukaryotes, and UvsX in viruses[96]. Outgroups for *nifH* consisted of sequences coding for the Pfam domain Fer4_NifH (PF00142), including a subunit of the pigment biosynthesis complexes protochlorophyllide reductase and chlorophyllide reductase[97].

**Phylogenetic analysis of recruited metagenomic reads.** To support the taxonomic affiliation of metagenomic reads recruited by *nifH* sequences from 'spheroid bodies', we carried out a phylogenetic reconstruction in the following way. The translated metagenomic reads were aligned against a NifH reference alignment using the option --add of MAFFT version 6 with the G-INS-I strategy[98]. The resulting protein alignment was used for aligning the corresponding nucleotide sequences using TranslatorX[99] and phylogenetic trees were generated using the HKY85 substitution model in PhyML version 3.0[100]. Four categories of rate variation were used. The starting tree was a BIONJ tree and the type of tree improvement was subtree pruning and regrafting. Branch support was calculated using the approximate likelihood ratio test (aLRT) with a Shimodaira–Hasegawa-like (SH-like) procedure.

**Flow cytometry data and analysis.** For quantifying the densities of single-cell free-living diazotrophs, we combined the cell density measurements from flow cytometry with the relative abundances derived from molecular methods[101]. Specifically, we multiplied the bacterial concentration derived from flow cytometry by the *nifH* to *recA* ratio of metagenomic read abundances from samples of size fraction 0.22–1.6/3 μm. For biovolume estimations of single-cell free-living diazotrophs, we assumed an average cell biovolume of 1 μm³ based on the cell dimensions reported in literature for cultured NCDs[102–107].

The free-living bacterial densities were retrieved from the last version of the publicly available flow cytometry *Tara* Oceans dataset (https://doi.org/10.17632/p9r9wttjkm.2), which were generated in the following way (see also refs. [23,108]). Briefly, three 1-mL seawater samples (pre-filtered through 200 μm mesh) were collected from the three layers (surface, DCM, and mesopelagic) using the vertical profile sampling system (CTD-rosette). Cold 25% glutaraldehyde (final concentration of 0.125%) was amended to each sample for 10 min, which was held in the dark prior to flash freezing in liquid nitrogen, and long-term storage at −80 °C until analyses. Two subsamples (400 μl) were stained with 1:10 SybrGreen 1 stock to a 2.5 μmol L$^{-1}$ final concentration for enumerating the heterotrophic prokaryotes in separate counts. After 10 min of staining in the dark, each sample was loaded into a FACsCalibur (Becton and Dickinson, Franklin Lakes, NJ, USA) flow cytometer fitted with a 15-mW Argon-ion laser (4888-nm emission). A summary of the gating strategy is shown in Supplementary Fig. S13. Two populations of heterotrophic prokaryotes are distinguished: high and low containing DNA, by the combination of side scatter (SSC), as well as nucleic acid derived SybrGreen green fluorescence (FL1). A known density of fluorescent beads (1 μm, Fluoresbrite carboxylate microspheres, Polysciences, Inc., Warrington, PA, USA) was used as an internal standard. The two other subsamples were not stained and used to enumerate phototrophic pico-phytoplankton, including *Prochlorococcus* and *Synechococcus*. The latter were discriminated by gating on side scatter, chlorophyll content (FL3, red fluorescence), and phycoerythrin (FL2, orange fluorescence). A minimum of 30,000 events and on average 90,000 events were acquired for each subsample. Total abundance of prokaryotic cells was based on the abundances of heterotrophic prokaryotes and phototrophic prokaryotes.

**Detection of diazotrophs in the confocal laser-scanning microscopy dataset.** We analyzed the quantitative microscopy images publicly available on the Ecotaxa web platform[109]. These images were generated using environmental High Content Fluorescence Microscopy (eHCFM)[30] on (i) 61 samples collected at 48 different stations using a microplankton net (20 μm mesh size) and filtering the cod end volume through a 180 μm sieve; (ii) 75 samples collected at 51 stations using a nanoplankton net (5 μ mesh size) and filtering the coded volume through a 20 μm sieve (Supplementary Fig. S2). Sample collection and preparation were, briefly, as follows: samples were fixed on board *Tara* in 10% monomeric formaldehyde (1 % final concentration) buffered at pH 7.5 and 500 μl EM grade glutaraldehyde (0.25% final concentration) and kept at 4 °C until analysis. Cells were imaged by Confocal Laser Scanning Microscopy (Leica Microsystem SP8, Leica Germany), equipped with several laser lines (405, 488, 552, 638 nm). The 5–20 μm size fraction was imaged with the water immersion lens HC PL APO 40x/1,10 mot CORR CS2 objective, and the 20–180 μm size fraction was imaged with the HC PL APO 20x/0.75IMM CORR CS2 objective. Multiple fluorescent dyes were used to observe the cellular components of the organisms, including the nuclei (blue, Hoechst, Ex405 nm/Em420–470 nm), cellular membranes (green, DiOC6(3), Ex488 nm/Em500–520 nm), cell surface (cyan, AlexaFluor 546, Ex552 nm/Em560–590 nm), and chlorophyll autofluorescence (red, Ex638 nm/Em680–700 nm).

We used the confocal microscopy data to quantify DDAs and free filaments of *Trichodesmium* for abundance and biovolume. Image classification and annotation were carried out using the Ecotaxa web platform[109] in the following way. We first manually searched for the target taxa and curated an initial training set in a few samples where molecular methods detected high abundances (i.e., high metagenomic read abundance of *nifH*), obtaining 55 images for DDAs and 39 for *Trichodesmium* filaments (Supplementary Data 5). This training set was then used for automating the classification of the whole dataset by means of supervised machine learning (random forest) based on a collection of 480 numeric 2D/3D features[30]. The predictions were, in turn, manually curated and used as a new training set, repeating this step iteratively until no new images appeared. Other taxonomic groups were also annotated and used as outgroups to improve the predictions of our taxa of interest. Abundance estimates were normalized based on the total sample volumes as cells L$^{-1}$ (see below). DDAs are enumerated as symbiotic cells L$^{-1}$, while *Trichodesmium* are filaments L$^{-1}$. We used the major and minor axis of every image to calculate their ellipsoidal equivalent biovolume.

**Ploidy calculation for *Trichodesmium* and *Richelia/Calothrix*.** For ploidy calculations, we used the paired confocal microscopy imaging and metagenomes obtained from the same 20–180-μm size-fractionated samples, thus dividing the cells L$^{-1}$ by the *nifH* copies L$^{-1}$. A known volume of seawater ($V_{seawater}$, L) was filtered and recorded through the plankton net (20 μm mesh). Contents of the cod end were filtered through a 180 μm sieve and diluted to 3 L. Forty-five milliliters of the diluted cod end was fixed with 5 ml formaldehyde (1% final) and glutaraldehyde (0.25% final)[30], and kept at 4 °C. A subsample volume ($V_i$, liter) was mounted into an optical multi-well plate for automated confocal imaging and adjusted to avoid the saturation of the well bottom. The well bottom area (86 mm²) was fully imaged. The organisms were segmented from the images, and then identified and counted. Hence, the estimate of the abundance (A) per liter of a given biological group in the original seawater is provided by A = #occurrence/($V_i \times Cf$); the concentration factor Cf = ($V_{seawater}/3$ L) × (45/50). The number of *Richelia/Calothrix* filaments per host and number cells per filament in *Trichodesmium* and *Richelia/Calothrix* were enumerated manually, thus obtaining the values of cells L$^{-1}$ in each sample.

For nucleic acid isolation, 1000 mL were sampled from the diluted cod end and filtered onto 47-mm diameter 10 μm pore-size polycarbonate membrane filters (one to four membrane filters, *n*, depending on the sample), and immediately flash-frozen. DNA extraction was performed from one membrane as described by Alberti et al.[24]. Of the extracted DNA (DNA$_{extracted}$, ng), a quantity was used for DNA library preparation and sequencing process using Illumina machines (Illumina, USA)[24]. Thus, we calculated the *nifH* copies L$^{-1}$ for each taxa using the number of reads mapping the *nifH* reference sequences of the taxon of interest (reads$_{nifHtaxon}$) among the total number of sequenced reads (reads$_{total}$). We assumed 100% efficiency of DNA extraction for *Trichodesmium* and *Richelia/Calothrix* and a *nifH* gene length of 750 bp (which implies $1.21 \times 10^9$ *nifH* copies/*nifH* ng). Hence, the estimate of the *nifH* copies (C) per liter of a given biological group in the original seawater is provided by C = DNA$_{extracted}$ * (1.21 × 10$^9$ copies *nifH*/ng *nifH*) * reads$_{nifHtaxon}$/reads$_{total}$/($V_{seawater}$/(3 L * n)).

**Underwater Vision Profiler dataset and analysis.** The Underwater Vision Profiler 5 (UVP5, Hydroptics, France)[33] is an underwater imager mounted on the Rosette Vertical Sampling System. This system illuminates precisely calibrated volumes of water and captures images during the descent (5–20 images s$^{-1}$). The UVP5 was operated in situ and was designed to detect and count objects of >100 μm in length and to identify those of >600 μm in size. In the current work, we used this method for the quantification of *Trichodesmium* colony abundance and biovolume by mining the images publicly available on the Ecotaxa web platform[109]. The search, curation and annotation of the corresponding images and their biovolume determination were carried out as described in the previous section. Only images clearly curated as tuft or puff colonies were used to avoid false positives

(fecal pellets, large diatom chains, etc). However, this stringent classification underestimated these colonies given the numerous stations without in situ UVP5 images annotated as *Trichodesmium* colonies but with detection of *Trichodesmium nifH* in the metagenomes of the 180–2000 μm size-fractionated samples (Fig. 6c). This pattern can also be produced if *Trichodesmium* colonies do not pass the filter holes of 2000 μm in diameter, but given that they are known to be fragile, it is probable that most of them pass it either undamaged or fragmented.

**Determination of contextual physico-chemical parameters.** Measurements of temperature were recorded at the time of sampling using the vertical profile sampling system (CTD-rosette) (https://doi.org/10.1594/PANGAEA.836319 and https://doi.org/10.1594/PANGAEA.836321). Dissolved nutrients (NO$^-_3$, PO$_4$) were analyzed according to previous methods[110,111] with detection limits of 0.05 and 0.02 μmol L$^{-1}$, respectively. Iron levels were derived from a global ocean biogeochemical model[112]. The datasets are listed in Supplementary Data 6.

**Plotting and statistical analysis.** Graphical analyses were carried out in R language (http://www.r-project.org/) using *ggplot2*[113] and treemaps were generated with *treemap*. Maps were generated with *borders* function in *ggplot2*[113] and *geom_point* function for bubbles or *scatterpie*[114] package for pie charts. The trends between diazotroph abundance and latitude were displayed with generalized additive models using the *geom_smooth* function of *ggplot2*[113]. Spearman's rho correlation coefficients and *p*-values were calculated using the *cor.test* function of the *stats* package. Metric multidimensional scaling (NMDS) analysis to visualize Bray–Curtis distances was carried out with the *metaMDS* command in the R package *vegan*[115], and the influence of environmental variables on sample ordination was evaluated with the function *envfit* in the same R package, whereas the pie charts were plotted with *scatterpie*[114] package. Hierarchical agglomerative clustering of samples using average linkage was performed with the function *hclust* of the R package *stats*.

**Reporting summary.** Further information on research design is available in the Nature Research Reporting Summary linked to this article.

## Data availability

The authors declare that all the data supporting the findings of this study are publicly available in the following repositories and in the supplementary information files of this paper. The contextual data are available in Pangaea (www.pangaea.de)[21] with the identifier https://doi.org/10.1594/PANGAEA.875582. Flow cytometry data[23,108] are available in Mendeley Data (https://data.mendeley.com) with the identifier https://doi.org/10.17632/p9r9wttjkm.2. The confocal microscopy[30] and UVP5[33] images annotated as diazotrophs in the current work and their corresponding metadata were submitted to the EMBL-EBI repository BioStudies (www.ebi.ac.uk/biostudies) under accession S-BSST529; while the whole image databases and their metadata are available at Ecotaxa (confocal microscopy of 5–20 μm sized-fractionated samples: https://ecotaxa.obs-vlfr.fr/prj/3365; confocal microscopy of 20–180 μm sized-fractionated samples: https://ecotaxa.obs-vlfr.fr/prj/2274; UVP5: https://ecotaxa.obs-vlfr.fr/prj/579). *Tara* Oceans metagenomes[22,24,25] are archived at ENA under the accession numbers: PRJEB1787, PRJEB1788, PRJEB4352, PRJEB4419, PRJEB9691, PRJEB9740, and PRJEB9742. The *nifH* and *recA* catalogs were compiled from Integrated Microbial Genomes (IMG, https://img.jgi.doe.gov/); Marine Microbial Eukaryotic Transcriptome Sequencing Project (MMETSP; https://github.com/dib-lab/dib-MMETSP); OM-Reference Gene Catalog version 2 (OM-RGC-v2, https://www.ocean-microbiome.org/); *Tara* Oceans assemblies (Supplementary Table 8 from ref. [16]); and the *nifH* database (version April 2014) curated and hosted at Zehr Lab, University of California, Santa Cruz, CA, USA (https://www.jzehrlab.com/nifh). The 10 *nifH* sequences generated in the clone libraries of the current work were submitted to ENA under the accession numbers: MW590317–MW590326. Source data are provided with this paper.

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

## Acknowledgements

We would like to thank all colleagues from the Tara Oceans consortium as well as the *Tara* Ocean Foundation for their inspirational vision. We also acknowledge the current and past members of the laboratory of JP Zehr for the curation and public access to the *nifH* database. We are grateful to Lionel Guidi, Stéphane Pesant, and Julie Poulain for providing details about *Tara* Oceans sampling, Luca Santangeli from EMBL Heidelberg for confocal microscopy image acquisition and Luis Pedro Coelho (currently at Fudan University) for image processing, Benjamin Blanc from Laboratoire d'Océanographie de Villefranche-sur-mer for help with UVP5 image processing and annotation, Josep Gasol from Institut de Ciències del Mar-CSIC for expertise and assistance with flow cytometry, Daniel Lundin from Linnaeus University for his analytical support and advice, Tom Delmont from Genoscope for useful discussions, Flora Vincent from Weizmann Institute

of Science for her help with Ecotaxa resource, Eric Bapteste, Philippe Lopez and Romain Lannes from Sorbonne Université for their advice with ultrasmall bacteria analysis. This work has been supported by the FFEM - French Facility for Global Environment, French Government 'Investissements d'Avenir' programs OCEANOMICS (ANR-11-BTBR-0008), FRANCE GENOMIQUE (ANR-10-INBS-09-08), MEMO LIFE (ANR-10-LABX-54), and PSL Research University (ANR-11-IDEX-0001-02). R.A.F. acknowledges funding from Knut and Alice Wallenberg foundation. C.B. acknowledges funding from the European Research Council (ERC) under the European Union's Horizon 2020 research and innovation program (Diatomic; grant agreement No. 835067) and Agence Nationale de la Recherche "Phytomet" (ANR-16-CE01-0008) projects. F.M.C.-C. acknowledges funding from the European Union's Horizon 2020 research program under the Marie Sklodowska-Curie grant agreement No. 749380 (UCYN2PLAST). S.G.A. acknowledges funding from the project "MAGGY" (CTM2017-87736-R) from the Spanish Ministry of Economy and Competitiveness and the 'Severo Ochoa Centre of Excellence' accreditation (CEX2019-000928-S). J.J.P.K. acknowledges postdoctoral funding from the Fonds Français pour l'Environnement Mondial. This article is contribution number 118 of *Tara* Oceans.

## Author contributions

R.A.F. and C.B. designed the study and supervised the project. R.A.F., C.B., and J.J.P.K. wrote the paper with substantial input from all co-authors. J.J.P.K. compiled the marker gene catalogs and E.P. and P.W. performed the metagenomic mapping. R.P., S.C., C.d.V., and E.K. set up the imaging platform for the eHCFM data generation and processing. J.J.P.K., R.A.F., M.C., E.D., F.L., and S.C. performed the taxonomic annotation of the eHCFM dataset. F.M.C.-C. and S.G.A. generated the *nifH* clone libraries. M.P. performed the collection and taxonomic annotation of UVP5 dataset. J.J.P.K. performed the formal analysis and visualization.

## Funding

## Competing interests

The authors declare no competing interests.
