## [Peer Review File · Nature Communications]

Reviewers' Comments:

Reviewer #1:

Remarks to the Author:

This manuscript describes a novel merger of imaging (confocal microscopy and UVP images of *Trichodesmium* colonies) with metagenomics and *nifH* markers. They found remarkable agreement between biogeographical implications via these disparate methods. Importantly they are able to take a biovolume approach to understanding which organisms or assemblages dominate in their database and the results conflict with prior gene based metrics of ucyn dominance. In short, this is an important contribution to our collective understanding of the biogeography and functional redundancy of diazotrophs in the global ocean. I do find the manuscript a bit extended for this journal given the length of the supplementary material and the main text/figures at the upper bounds of the journals editorial limits, but I believe it should be published. I have a number of comments of varying importance which I itemize below. Most importantly, I feel it unwarranted to speculate that these results "highlights new sources and sinks of diazotroph-fueled new production" as the data presented cannot be framed in the context of rate measurements. I also have some questions as to what the inherent error related to the flow cytometry/gene-based assessments of the NCD biomass are. Otherwise, most of my concerns should be readily addressed.

Specific comments are as follows:

- In the abstract the authors state that biological nitrogen fixation sustains "~50% of ocean primary productivity". This may be true in a *Trichodesmium* bloom or some hot-spot of N₂ fixation but generally it is reported that nitrogen fixation supports ~50% of export (~equivalent to net community production) and these are constrained to areas such as the subtropical gyres. Given e-ratios of 10-20% this would indicate nitrogen fixation supporting a far lower fraction of primary production (gross or net).

References:

<https://www.nature.com/articles/s41586-019-0911-2>; <https://www.nature.com/articles/41474>

- Abstract line 33: (0-2000). This should be changed to the true lower particle size limit. One cannot measure the infinitesimally small zero
- Abstract line 34 and throughout. I would suggest that the authors clarify when *nifH* sequences are detected, images or the combination thereof rather than 'diazotrophs'. I understand the sentiment but the issue of polyploidy, extraction efficiency, detection efficiency via imaging, and other issues are well documented in the literature such that *nifH* does not always align with imaging-based abundances. Clarifying what was detected versus 'diazotrophs' might make this distinction clearer and allow the reader to better understand what data are being presented.
- Abstract line 38. The authors might clarify if unicellular diazotrophs refer to cyanobacterial or cyanobacterial + non-cyanobacterial heterotrophs
- Abstract line 42. While this study clearly highlights biogeography of diazotrophs, I see no data highlight relative sources of N or sinks of 'diazotroph-fueled new production'
- Results line 120: Since the 61 samples were all in the 20-180 micron range, I would anticipate they would potentially underestimate the abundance of free filaments of *Trichodesmium* since the width of filaments is < 10 micron. The only publication reporting *Trichodesmium* filament morphology I am aware of reports "a mean filament length of 90µm and a range of 12-6801µm" (doi: 10.3389/fmars.2018.00027). The authors might consider a brief statement on the potential for underestimation of smaller filaments by use of a 20 micron filtration. Were *Trichodesmium* detected in the < 20 micron fraction of the metagenomic dataset?
- Line 135: Were heterocysts and vegetative cells both counted as "symbiotic cells"?
- Line 158: Might clarify that Station ALOHA is in the NPSG for any readers unaware
- Line 172: Given the low number of stations with observable colonies in Fig S4A-lower (and the total colonies as ~ 1K as shown in 2A), it seems it would be possible to add a link to the ecotaxa classifier of these colonies or generate a collage of validated colonies. I mention this only because in searching ecotaxa for this project it does not seem as if ~1K images were validated as *Trichodesmium* colonies (<https://ecotaxa.obs-vlfr.fr/prj/579>). Adding a clear view of the morphological variation of this validated classifier might be helpful for novices to this technology
- Line 185: I would clarify that you are using the combination of FCM, UVP, and microscopy instead of "extensive imaging" and might also include the number of images for each technology since there were orders of magnitude of variation

- Line 250: Is there any evidence for polyploidy in Richelia or Calothrix?
- Line 259: You might consider re-wording this statement. Also, if UCYN-A are in the larger fraction can this not also be in marine snow or as aggregates versus in the guts of consumers which is what I think you are referring to here?
- I am not convinced of the significance of 'spheroid bodies' which are only found in a single fraction and at very low levels. Is there any other interpretation for Fig S7 ? perhaps not, but it just seems like a detour in the text
- Line 302 – Could this alternately imply inaccuracy of modeled iron? I would add something like "assuming modeled iron concentrations are representative of natural concentrations" ...
- Line 311, I would clarify DDA instead of Richelia since the symbioses are exported, I would also use the Sharek reference in lieu of Karl et al. which only infers export from d15N and pSi flux
- Line 313. The same Scharek 1999 DSR paper on export cited herein records Trichodesmium export of <100- 120,540 cells/m²/d which is at or higher than some diatom fluxes. This counters the statement made on line 313
- Line 400/468. Is particle association/morphology really "life-history"? Perhaps referring to the functional redundancy of the diazotrophs is more appropriate.
- Line 449. I generally disagree that this work is "directly relevant to predicting C and N sources/sinks". As the authors note, these organisms are generally rare and 'patchy' and presence of diazotrophs cannot be directly related to either C or N export nor the magnitude of N added to an ecosystem. This is unnecessary speculation in my opinion.
- Line 460. Could subduction, downwelling or other physical processes be responsible for these observations in lieu of a 1-D conceptualization of export?
- Line 492 .."were also collected"
- Line 560. I am a bit confused by this method. If you calculate % bacterial diazotrophs as nifH/recA * abundance and then assume a fixed biovolume for the diazotrophs, what biovolume is assumed for the range of heterotrophic cells assayed by FCM? It also seems that some error analysis is warranted in this approach.
- Line 611. I would suggest adding the detection limit for the ISUS assuming this was calibrated by discrete measurements.
- Line 650 – contribution number needs to be added
- Line 697/736-737/760/822 and elsewhere. Some references needed to be checked for formatting and typographical errors
- Figure 1. This is a beautiful figure.
- Fig 2 It also seems counterintuitive to me that the geography of filaments and colonies of Tricho seem so different in Fig 2. The authors suggested this was due to local nutrient limitation- do the in situ ISUS nitrate support this?
- Fig 6. I am a bit amazed that filaments (which have a widely varying # of cells/filaments would so closely match the % tricho nifH/recA given known polyploidy and filament length. Since one could determine cells/filament or cells/L and there were only 61 samples, I am surprised this was not done. Does it improve or degrade the correlation or balance polyploidy? Or does it imply a stable filament length in the confocal microscopy? I have a similar comment with Richelia/Calothrix ... How much of this is driven by polyploidy or changes in length of vegetative cells?

Reviewer #2:

Remarks to the Author:

Karlusich et al. nicely leverage novel microscopic image data with extensive metagenomic data to improve resolution on diazotroph biogeography in the ocean.

The most exciting findings in my opinion are: 1) detection of 'hot spots' in understudied regions – which will advance the general research field by bringing attention to these areas and global N-fixation model tuning and 2) the detection of less-studied diazotroph populations and use of infrequently used size-fractionation sampling (<0.2 um) and novel imagery data to reveal basic knowledge of their ecology.

The findings are clearly reported in the text and will be of interest to oceanographers as well as microbial/plankton ecologists. That said the text is a bit long; the authors should consider slimming the Introduction and the Results and Discussion in areas where the data confirms prior

observations/paradigms.

The methods used appear sound – esp. use of non-parametric statistical analyses.

Overall I have several specific comments/suggested edits that I would like to see addressed before this manuscript is ready for publication:

Line 28: N-fixation sustains 50% of ocean primary production (PP)? About 50% of PP occurs in coastal waters where N-fixation is very low. This statement is not true; at times in select oligotrophic regions this can be true. Perhaps the intention was to refer to 'new' PP?

Line 33: $<0.22 - 2000\mu\text{m}$ makes more sense to me... no cells are $0\mu\text{m}$.

Line 40: 'complex overlapping niches' ... this is a little unclear; are you meaning there are likely subtly different niches that prevent competitive exclusion?

Line 50: On geologic time scales this statement is true.

Line 62: Lineages rather than strains.

Line 65: Name the single diatom spp.

Line 69: Archaeal, rather than Archaea

Lines 74 – 88: The year ranges provided are distracting; I'm not seeing why they need to be included.

Line 104-107: This could be clearer... specifically that you are analyzing the abundance of the nifH gene normalized to recA abundance. Also, there should be a citation related prior to use of recA as a single copy gene and caveats.

Line 141: It may not be obvious to general readers how one determines that the cell is compromised. What was the criteria?

Line 161 – 170: This section could be more concise. Why is exactly was it unexpected to find Trichodesmium in the Pacific N. Eq. Current?

Line 172: the NAO

Line 175: Is there no nutrient data to back up this claim?

Line 191: Here an arbitrary volume of $1\mu\text{m}^3$ is picked for NCD calculations. Why not reference a paper that has measured the size/volume of an NCD isolate or give a better argument as to why $1\mu\text{m}^3$ is reasonable? I doubt the NCD contribution will begin to approach that of Trichodesmium with slightly higher biovolume, but this statement as is leaves uncertainty on the issue.

Lines 243: The end of this sentence is not grammatically correct. Also grazing deserves consideration here – not just attachment to larger plankton.

Line 251-253: It seems like you should be able to address this polyploidy issue with the data you have by comparing Trichodesmium cells (within filaments) in the imagery data and nifH sequence numbers. Assuming equivalent extraction efficiency across diazotrophs – if there is notably higher polyploidy in Trichodesmium I'd expect the slope of a cell # to nifH # be quite different from other taxa?

Line 258: Grazing needs to be considered here.

Line 276: Remove 'heretofore'.

Line 335: There could also be more efficient aggregation/grazing/export rather than a surface “bloom” event.

Line 377-378: This last sentence is confusing and does not seem necessary. The most abundant plankton in the ocean are not very patchy at a broad view – e.g. SAR11, Prochlorococcus spp. Rarer plankton are much patchier.

Line 435-441: As I understand it, the percent abundance of these ‘ultrasmall diazotrophs’ is based on nifH/recA abundance, correct? Are there no estimates of ultraplankton abundance? ‘Fundamental revisit of marine nitrogen fixation’ is very vague – please clarify. It is possible to make some ‘back of the envelope’ calculations regarding the potential impact of these tiny diazotrophs based on activity of slightly larger diazotrophs... this would be more convincing that these cells need to be considered in global (or Arctic?) N-cycling models.

Line 445: remove the parentheses around ‘PCR-free’. Also consider mentioning this somewhere in the abstract – this is important given the issues of PCR bias and the historically heavy reliance on PCR to study diazotrophs.

Line 467: This sentence could be clearer. I’m not seeing a new paradigm shift here – I think the marine diazotroph research community already recognizes that these different populations occupy different niches and have different life histories.

Line 534: There is a typographical error in this sentence.

Line 650: Typo error.

Line 697: Capitalization error... check all references.

Figure 8. The font along the y-axis is very small; there are a lot of stations to display here but could it be made clearer? Possibly with an angled legend?

POINT-BY-POINT RESPONSE TO THE REVIEWER' COMMENTS

Reviewer #1:

This manuscript describes a novel merger of imaging (confocal microscopy and UVP images of *Trichodesmium* colonies) with metagenomics and *nifH* markers. They found remarkable agreement between biogeographical implications via these disparate methods. Importantly they are able to take a biovolume approach to understanding which organisms or assemblages dominate in their database and the results conflict with prior gene based metrics of ucyn dominance. In short, this is an important contribution to our collective understanding of the biogeography and functional redundancy of diazotrophs in the global ocean. I do find the manuscript a bit extended for this journal given the length of the supplementary material and the main text/figures at the upper bounds of the journal editorial limits, but I believe it should be published. I have a number of comments of varying importance which I itemize below. Most importantly, I feel it unwarranted to speculate that these results “highlights new sources and sinks of diazotroph-fueled new production” as the data presented cannot be framed in the context of rate measurements. I also have some questions as to what the inherent error related to the flow cytometry/gene-based assessments of the NCD biomass are. Otherwise, most of my concerns should be readily addressed.

REPLY: We are pleased to know that the manuscript was well received, and we thank the reviewer for her/his helpful comments and corrections. We address them below.

Specific comments are as follows:

- In the abstract the authors state that biological nitrogen fixation sustains “~50% of ocean primary productivity”. This may be true in a *Trichodesmium* bloom or some hot-spot of N₂ fixation but generally it is reported that nitrogen fixation supports ~50% of export (~equivalent to net community production) and these are constrained to areas such as the subtropical gyres. Given e-ratios of 10-20% this would indicate nitrogen fixation supporting a far lower fraction of primary production (gross or net).

References:

<https://www.nature.com/articles/s41586-019-0911-2>; <https://www.nature.com/articles/41474>

REPLY: We agree with the correction, and have modified the abstract accordingly (line 31).

- Abstract line 33: (0-2000). This should be changed to the true lower particle size limit. One cannot measure the infinitesimally small zero

REPLY: Corrected (line 36)

- Abstract line 34 and throughout. I would suggest that the authors clarify when *nifH* sequences are detected, images or the combination thereof rather than ‘diazotrophs’. I understand the sentiment but the issue of polyploidy, extraction efficiency, detection efficiency via imaging, and other issues are well documented in the literature such that *nifH* does not always align with imaging-based abundances. Clarifying what was detected versus ‘diazotrophs’ might make this distinction clearer and allow the reader to better understand what data are being presented.

REPLY: We agree this was not clear and thus modified the text accordingly (lines 37-44).

• Abstract line 38. The authors might clarify if unicellular diazotrophs refer to cyanobacterial or cyanobacterial + non-cyanobacterial heterotrophs

REPLY: We adjusted the text to clarify which diazotrophic groups (lines 43-44).

• Abstract line 42. While this study clearly highlights biogeography of diazotrophs, I see no data highlight relative sources of N or sinks of 'diazotroph-fueled new production'

REPLY: We agree this was not an appropriate interpretation and removed it.

• Results line 120: Since the 61 samples were all in the 20-180 micron range, I would anticipate they would potentially underestimate the abundance of free filaments of *Trichodesmium* since the width of filaments is < 10 micron. The only publication reporting *Trichodesmium* filament morphology I am aware of reports "a mean filament length of 90 μm and a range of 12–6801 μm " (doi: 10.3389/fmars.2018.00027). The authors might consider a brief statement on the potential for underestimation of smaller filaments by use of a 20 micron filtration. Were *Trichodesmium* detected in the < 20 micron fraction of the metagenomic dataset?

REPLY: Indeed *Trichodesmium* spp. *nifH* sequences were detected in the metagenomes of the 0.8-5 μm , and to a greater extent in the 5-20 μm , and by far the highest detection comes from the largest size fractions (20-180 μm and 180-2000 μm) (see Figures 5c and 8). In addition to the results presented based on searches in the confocal microscopy images of the 20-180 size fraction, we carried out the same search in the confocal microscopy images from the 5-20 μm size fraction (Colin et al 2017). The new analyses included 75 samples from 51 stations from tropical and subtropical regions, and resulted in detection of only 8 *Trichodesmium* filament images, corresponding to 0.2-0.8 filaments L^{-1} . Therefore, we discarded a potential underestimation of smaller filaments by use of a the plankton net of 20 μm mesh. However, it is important to note that among the sampled filaments in the 20-180 μm size fraction, 35% of them are >180 μm in length, suggesting that there is a potential underestimation of long filaments due to the filtering with the 180 μm sieve. We have added this information in the text (lines 129-134 and 173-175).

• Line 135: Were heterocysts and vegetative cells both counted as "symbiotic cells"?

REPLY: Each diatom host cell harboring *Richelia/Calothrix* filaments was counted as a "symbiotic cell" (line 148). In addition, the number of total cells (both heterocysts and vegetative ones) per filament were enumerated manually as this was necessary for the ploidy calculations (see below).

• Line 158: Might clarify that Station ALOHA is in the NPSG for any readers unaware

REPLY: We have added the appropriate information and defined the acronyms (lines 166-167).

• Line 172: Given the low number of stations with observable colonies in Fig S4A-lower (and the total colonies as ~ 1K as shown in 2A), it seems it would be possible to add a link to the ecotaxa classifier of these colonies or generate a collage of validated colonies. I mention this only because in searching ecotaxa for this project it does not seem as if ~1K images were validated as *Trichodesmium* colonies (<https://ecotaxa.obs-vlfr.fr/prj/579>). Adding a clear view of the morphological variation of this validated classifier might be helpful for novices to this technology

REPLY: The values in Fig 2A and Supplementary Fig. S4 corresponds to the concentrations of colonies, which are calculated using the total volume covered by UVP when a given image was detected. A total of 220 UVP images were used in our work, corresponding to only images that clearly curated as tuft or puff colonies and thus avoiding false positives (faecal pellets, large diatom chains, etc) (we added this information in lines 179-187). All 220 images are deposited and fully available in <https://ecotaxa.obs-vlfr.fr/prj/579> under the annotation categories 'tuft' and 'puff'. In addition, they were submitted to the EMBL-EBI repository BioStudies (www.ebi.ac.uk/biostudies) under accession S-BSST529, as stated in the Reporting Summary and now also in the Data availability section. We have also added the link in the corresponding Results section for easy access to the readers (line 185).

- Line 185: I would clarify that you are using the combination of FCM, UVP, and microscopy instead of "extensive imaging" and might also include the number of images for each technology since there were orders of magnitude of variation

REPLY: We have clarified the combination of image analyses (lines 200-201) and have added the number of images using each technology in the appropriate section of results (lines 125-126, and 183-185).

- Line 250: Is there any evidence for polyploidy in *Richelia* or *Calothrix*?

REPLY: To our knowledge there are no studies on polyploidy in *Richelia* and *Calothrix*, only on related heterocystous cyanobacteria (Sukenik et al. 2011. *IsmeJ*. 6: 670-679). We have added details and a new section on ploidy in the results (lines 299-328).

- Line 259: You might consider re-wording this statement. Also, if UCYN-A are in the larger fraction can this not also be in marine snow or as aggregates versus in the guts of consumers which is what I think you are referring to here?

REPLY: We agree with this suggestion and have modified the text accordingly (lines 272-275).

- I am not convinced of the significance of 'spheroid bodies' which are only found in a single fraction and at very low levels. Is there any other interpretation for Fig S7 ? perhaps not, but it just seems like a detour in the text

REPLY: We agree that these results are less robust compared to the other datasets. However, we identified the *nifH* gene for these 'spheroid bodies' in samples from 3 distant ocean basins and in a size fraction consistent with current information. More specifically, based on the genome content of these 'spheroid bodies', they are considered obligate (Nakayama et al. 2014 *PNAS* 111: 11407-11412), thus they should only be detected in a size fraction that would collect their respective hosts. We have shortened this section (lines 280-297) and provided more explanations as to how we attempted to identify these symbioses by image recognition. We plan to do a deeper analysis that can be the scope of a future work pointing to the existence of 'spheroid bodies' in the ocean.

- Line 302 – Could this alternately imply inaccuracy of modeled iron? I would add something like "assuming modeled iron concentrations are representative of natural concentrations" ...

REPLY: We agree with this point and added an explanation (lines 341-342).

• Line 311, I would clarify DDA instead of Richelia since the symbioses are exported, I would also use the Sharek reference in lieu of Karl et al. which only infers export from d15N and pSi flux

REPLY: We agree and have modified accordingly and added the two Scharek references: Scharek et al 1999 MEPS 182: 55-67; Scharek et al 1999 Deep Sea Res I 46: 1051-1076 (lines 351-353).

• Line 313. The same Scharek 1999 DSR paper on export cited herein records *Trichodesmium* export of <100- 120,540 cells/m²/d which is at or higher than some diatom fluxes. This counters the statement made on line 313

REPLY: We have modified these statements to include *Trichodesmium* as a significant contributor to export (lines 353-355).

• Line 400/468. Is particle association/morphology really “life-history”? Perhaps referring to the functional redundancy of the diazotrophs is more appropriate.

REPLY: We have replaced here and elsewhere ‘life-history’ with “lifestyle” (lines 45, 103, 277, 357, 436, 503).

• Line 449. I generally disagree that this work is “directly relevant to predicting C and N sources/sinks”. As the authors note, these organisms are generally rare and ‘patchy’ and presence of diazotrophs cannot be directly related to either C or N export nor the magnitude of N added to an ecosystem. This is unnecessary speculation in my opinion.

REPLY: We acknowledge this statement was speculative and have modified. We highlight that we have identified new regions (‘hotspots’) for future studies (lines 494-496).

• Line 460. Could subduction, downwelling or other physical processes be responsible for these observations in lieu of a 1-D conceptualization of export?

REPLY: We have now acknowledged that physical processes could indeed influence the results, and have modified accordingly (lines 378-381 and 490-494).

• Line 492 ..”were also collected”

REPLY: Corrected (lines 524-525).

• Line 560. I am a bit confused by this method. If you calculate % bacterial diazotrophs as nifH/recA * abundance and then assume a fixed biovolume for the diazotrophs, what biovolume is assumed for the range of heterotrophic cells assayed by FCM? It also seems that some error analysis is warranted in this approach.

REPLY: We have now defined a consensus cell size based on values reported in the literature for marine and non-marine NCDs (lines 605-608). These included the gammaproteobacteria *Marinobacterium mangrovicola* (Alfaro-Espinoza and Ullrich 2014 Int J Syst Evol Microbiol 64.12: 3988-3993), *Teredinibacter turnerae* (Distel et al. 2002 Int J Syst Evol Microbiol 52.6: 2261-2269), *Azotobacter vinelandii* (Inomura et al 2017 ISME J 11: 166–175), *Pseudomonas stutzeri* (Lalucat et al. 2006 Microbiol Mol Biol Rev 70: 510–547), *Cobetia* spp (Romanenko et al 2013 Int. J. Syst. Evol. Microbiol. 63, 288–297), and the alphaproteobacterium *Rhodopseudomonas paropalustris* (Ramana et al 2012 Int J Syst Evol Microbiol 62:1790-1798) .

• Line 611. I would suggest adding the detection limit for the ISUS assuming this was calibrated by discrete measurements.

REPLY: The detection limits were added for the nitrate (0.05 μM) and phosphate (0.02 μM) concentrations (lines 700-703).

• Line 650 – contribution number needs to be added

REPLY: Thank you for pointing this out; it will be added if the paper is accepted at the proof stage (line 756).

• Line 697/736-737/760/822 and elsewhere. Some references needed to be checked for formatting and typographical errors

REPLY: References were checked and corrected.

• Figure 1. This is a beautiful figure.

REPLY: We are pleased to know this.

• Fig 2 It also seems counterintuitive to me that the geography of filaments and colonies of *Tricho* seem so different in Fig 2. The authors suggested this was due to local nutrient limitation- do the in situ ISUS nitrate support this?

REPLY: We added the plots with the distribution of free filaments and colonies according to nutrients (Supplementary Fig. S6), which suggest high abundance of colonies with respect to free filaments in low concentrations of iron, phosphate and nitrate. These field patterns support previous results under culture conditions (Tzubarí et al. 2018 ISME J 12: 1682–1693). Lines 187-191.

• Fig 6. I am a bit amazed that filaments (which have a widely varying # of cells/filaments would so closely match the % tricho nifH/recA given known polyploidy and filament length. Since one could determine cells/filament or cells/L and there were only 61 samples, I am surprised this was not done. Does it improve or degrade the correlation or balance polyploidy? Or does it imply a stable filament length in the confocal microscopy? I have a similar comment with *Richelia/Calothrix* ... How much of this is driven by polyploidy or changes in length of vegetative cells?

REPLY: A new section on polyploidy was added in the results (lines 299-328). However, it is important to note that the metagenomic sampling from *Tara* Oceans was not specifically designed to quantify metagenomic signals per seawater volume due to the lack of ‘spike-ins’ or DNA internal standards, and thus we have made assumptions (i.g. DNA extraction is 100%) in order to approximate estimates using the reported sampled seawater volumes and the quantity of extracted DNA.

Briefly, based on the assumptions, we estimate that polyploidy for *Trichodesmium* is 410 (geometric mean of 15 points) and 177 for *Richelia/Calothrix* (geometric mean of 21 points), and both taxa show 5 orders of magnitude in the variability of polyploidy values according to the sample. This is summarized in the text along with comparison to literature for *Trichodesmium* (e.g. range of 1-600; Sargent et al 2016 FEMS Microbiol Lett 363, fnw244). The average *Trichodesmium* filament length from our analyses was 155 μm , and the average number of *Richelia* cells filament depended on host such that *Richelia* associated with *Hemialus* were 5 and *Rhizosolenia* were 8 and *Calothrix* associated with *Chaetoceros* were 5. We choose to limit our speculations since the observational data cannot be used to interpret the various factors (e.g. growth conditions, nutrient status, developmental stage and

cell cycle) known to influence polyploidy (Maldonado et al 1994 J Bacteriol 176: 3911–3919; Simon 1977 J Bacteriol 129: 1154–1155; Simon 1980 FEMS Microbiol Lett 8: 241–245; Griese et al 2011 FEMS Microbiol Lett 323: 124–131; Sukenik et al 2012 ISME J 6: 670–679; Sargent et al 2016 FEMS Microbiol Lett 363, fnw244).

Reviewer #2:

Karlusich et al. nicely leverage novel microscopic image data with extensive metagenomic data to improve resolution on diazotroph biogeography in the ocean.

The most exciting findings in my opinion are: 1) detection of ‘hot spots’ in understudied regions – which will advance the general research field by bringing attention to these areas and global N-fixation model tuning and 2) the detection of less-studied diazotroph populations and use of infrequently used size-fractionation sampling (<0.2 μm) and novel imagery data to reveal basic knowledge of their ecology.

The findings are clearly reported in the text and will be of interest to oceanographers as well as microbial/plankton ecologists. That said the text is a bit long; the authors should consider slimming the Introduction and the Results and Discussion in areas where the data confirms prior observations/paradigms.

The methods used appear sound – esp. use of non-parametric statistical analyses.

Overall I have several specific comments/suggested edits that I would like to see addressed before this manuscript is ready for publication:

REPLY: We are grateful to Reviewer #2 for her/his encouraging comments. In the following lines we offer a reply to her/his helpful suggestions.

Line 28: N-fixation sustains 50% of ocean primary production (PP)? About 50% of PP occurs in coastal waters where N-fixation is very low. This statement is not true; at times in select oligotrophic regions this can be true. Perhaps the intention was to refer to ‘new’ PP?

REPLY: We agree and have modified our abstract accordingly (line 31).

Line 33: <0.22 – 2000 μm makes more sense to me... no cells are 0 μm .

REPLY: Corrected (line 36).

Line 40: ‘complex overlapping niches’ ... this is a little unclear; are you meaning there are likely subtly different niches that prevent competitive exclusion?

REPLY: We agree this was not clear and thus we removed it.

Line 50: On geologic time scales this statement is true.

REPLY: Corrected (lines 55-59).

Line 62: Lineages rather than strains.

REPLY: Corrected (line 70).

Line 65: Name the single diatom spp.

REPLY: Added (line 73-74).

Line 69: Archaeal, rather than Archaea

REPLY: Corrected (line 77).

Lines 74 – 88: The year ranges provided are distracting; I'm not seeing why they need to be included.

REPLY: The year ranges were removed and in addition we simplified the whole paragraph (lines 82-90).

Line 104-107: This could be clearer... specifically that you are analyzing the abundance of the *nifH* gene normalized to *recA* abundance. Also, there should be a citation related prior to use of *recA* as a single copy gene and caveats.

REPLY: We added an explanation about the use of *recA* as well as references that *recA* is a single copy gene and the previous citations that have used *recA* as a marker gene (lines 109-112).

Line 141: It may not be obvious to general readers how one determines that the cell is compromised. What was the criteria?

REPLY: We added in the text that we observed many broken chains/frustules with poor chlorophyll auto-fluorescence in the host diatoms, as well as short filaments of *Richelia*. Lines 152-154.

Line 161 – 170: This section could be more concise. Why is exactly was it unexpected to find *Trichodesmium* in the Pacific N. Eq. Current?

REPLY: The section was modified and removed the unexpected observation (lines 173-191).

Line 172: the NAO

REPLY: Corrected (line 188).

Line 175: Is there no nutrient data to back up this claim?

REPLY: We added the plots with the distribution of free filaments and colonies according to nutrients (Supplementary Fig. S6), which suggest high abundance of colonies with respect to free filaments in low concentrations of iron, phosphate and nitrate. These field patterns support previous results under culture conditions (Tzubarí et al. 2018 ISME J 12: 1682–1693). Lines 187-191.

Line 191: Here an arbitrary volume of 1 μm^3 is picked for NCD calculations. Why not reference a paper that has measured the size/volume of an NCD isolate or give a better argument as to why 1 μm^3 is reasonable? I doubt the NCD contribution will begin to approach that of *Trichodesmium* with slightly higher biovolume, but this statement as is leaves uncertainty on the issue.

REPLY: We have now defined a consensus cell size based on values reported in the literature for marine and non-marine NCDs. These included the gammaproteobacteria *Marinobacterium mangrovicola* (Alfaro-Espinoza and Ullrich 2014 Int J Syst Evol Microbiol 64.12: 3988-3993), *Teredinibacter turnerae* (Distel et al. 2002 Int J Syst Evol Microbiol 52.6: 2261-2269), *Azotobacter vinelandii* (Inomura et al 2017 ISME J 11: 166–175), *Pseudomonas stutzeri* (Lalucat et al. 2006 Microbiol Mol Biol Rev 70: 510–547), *Cobetia* spp (Romanenko

et al 2013 Int. J. Syst. Evol. Microbiol. 63, 288–297), and the alphaproteobacterium *Rhodopseudomonas paraparalustris* (Ramana et al 2012 Int J Syst Evol Microbiol 62:1790-1798). Lines 605-608.

Lines 243: The end of this sentence is not grammatically correct. Also grazing deserves consideration here – not just attachment to larger plankton.

REPLY: Corrected (lines 254-258).

Line 251-253: It seems like you should be able to address this polyploidy issue with the data you have by comparing *Trichodesmium* cells (within filaments) in the imagery data and *nifH* sequence numbers. Assuming equivalent extraction efficiency across diazotrophs – if there is notably higher polyploidy in *Trichodesmium* I'd expect the slope of a cell # to *nifH* # be quite different from other taxa?

REPLY: A new section on polyploidy was added in the results (lines 299-328). However, it is important to note that the metagenomic sampling from *Tara* Oceans was not specifically designed to quantify metagenomic signals per seawater volume due to the lack of 'spike-ins' or DNA internal standards, and thus we have made assumptions (i.g. DNA extraction is 100%) in order to approximate estimates using the reported sampled seawater volumes and the quantity of extracted DNA.

Briefly, based on the assumptions, we estimate that polyploidy for *Trichodesmium* is 410 (geometric mean of 15 points) and 177 for *Richelia/Calothrix* (geometric mean of 21 points), and both taxa show 5 orders of magnitude in the variability of polyploidy values according to the sample. This is summarized in the text along with comparison to literature for *Trichodesmium* (e.g. range of 1-600; Sargent et al 2016 FEMS Microbiol Lett 363, fnw244).

Line 258: Grazing needs to be considered here.

REPLY: We reworded the sentence to clarify that we are referring to grazing (line 272-275).

Line 276: Remove 'heretofore'.

REPLY: Done

Line 335: There could also be more efficient aggregation/grazing/export rather than a surface "bloom" event.

REPLY: We have added this alternative explanation (lines 373-381 and 490-494).

Line 377-378: This last sentence is confusing and does not seem necessary. The most abundant plankton in the ocean are not very patchy at a broad view – e.g. SAR11, *Prochlorococcus* spp. Rarer plankton are much patchier.

REPLY: This has been corrected (lines 418-420).

Line 435-441: As I understand it, the percent abundance of these 'ultrasmall diazotrophs' is based on *nifH/recA* abundance, correct? Are there no estimates of ultraplankton abundance? 'Fundamental revisit of marine nitrogen fixation' is very vague – please clarify. It is possible to make some 'back of the envelope' calculations regarding the potential impact of these tiny diazotrophs based on activity of slightly larger diazotrophs... this would be more convincing that these cells need to be considered in global (or Arctic?) N-cycling models.

REPLY: The reviewer is correct, the abundance percentage of ultrasmall diazotrophs is based on *nifH/recA* ratio in the metagenomes. Unfortunately, we do not have estimates for

absolute abundances of ultrasmall plankton, so we have reworded the text accordingly (lines 472-474).

Line 445: remove the parentheses around 'PCR-free'. Also consider mentioning this somewhere in the abstract – this is important given the issues of PCR bias and the historically heavy reliance on PCR to study diazotrophs.

REPLY: We appreciate this suggestion. This has now been modified in the discussion (line 477-479) and we also added it in the abstract (line 36).

Line 467: This sentence could be clearer. I'm not seeing a new paradigm shift here – I think the marine diazotroph research community already recognizes that these different populations occupy different niches and have different life histories.

REPLY: This has been modified. The paradigm shift is that rather than one dominant diazotroph (*Trichodesmium* vs. UCYN vs. NCDs or the >10 um vs. <10um), we present the concept of co-occurrence since there were several regions in which diazotroph co-occurrence was more common than one dominant group. We have tried to present this in a more clear way. Lines 499-504.

Line 534: There is a typographical error in this sentence.

REPLY: Corrected (line 541).

Line 650: Typo error.

REPLY: Contribution number will be added, if the paper is accepted, at the proof stage. Line 756.

Line 697: Capitalization error... check all references.

REPLY: References were checked and corrected.

Figure 8. The font along the y-axis is very small; there are a lot of stations to display here but could it be made clearer? Possibly with an angled legend?

REPLY: The figure was modified accordingly.

Reviewers' Comments:

Reviewer #1:

Remarks to the Author:

The authors have well addressed all of my concerns and I recommend publication.

Reviewer #2:

Remarks to the Author:

All of my questions/comments were addressed with the provided revisions. Well done by the authors.

I have only one other suggested edit in the Supplemental information: In Supplemental Figure S12 change 0-0.2 to <0.2 along the x-axes.

REVIEWERS' COMMENTS

Reviewer #1 (Remarks to the Author):

The authors have well addressed all of my concerns and I recommend publication.

REPLY: We are very pleased to hear this, and we appreciate the reviewer's helpful comments and corrections.

Reviewer #2 (Remarks to the Author):

All of my questions/comments were addressed with the provided revisions. Well done by the authors.

REPLY: We are very pleased to read this, and we thank the reviewer again for her/his helpful comments and corrections.

I have only one other suggested edit in the Supplemental information:

In Supplemental Figure S12 change 0-0.2 to <0.2 along the x-axes.

Done